# SEM-2/SoxC regulates multiple aspects of *C. elegans* postembryonic mesoderm development

**Marissa Baccas, Vanathi Ganesan, Amy Leung, Lucas R. Pineiro, Alexandra N. McKillop, Jun Liu** *

Department of Molecular Biology and Genetics, Cornell University, Ithaca, New York, United States of America

* kelly.jun.liu@cornell.edu

## Abstract

Development of multicellular organisms requires well-orchestrated interplay between cell-intrinsic transcription factors and cell-cell signaling. One set of highly conserved transcription factors that plays diverse roles in development is the SoxC group. *C. elegans* contains a sole SoxC protein, SEM-2. SEM-2 is essential for embryonic development, and for specifying the sex myoblast (SM) fate in the postembryonic mesoderm, the M lineage. We have identified a novel partial loss-of-function *sem-2* allele that has a proline to serine change in the C-terminal tail of the highly conserved DNA-binding domain. Detailed analyses of mutant animals harboring this point mutation uncovered new functions of SEM-2 in the M lineage. First, SEM-2 functions antagonistically with LET-381, the sole *C. elegans* FoxF/C forkhead transcription factor, to regulate dorsoventral patterning of the M lineage. Second, in addition to specifying the SM fate, SEM-2 is essential for the proliferation and diversification of the SM lineage. Finally, SEM-2 appears to directly regulate the expression of *hlh-8*, which encodes a basic helix-loop-helix Twist transcription factor and plays critical roles in proper patterning of the M lineage. Our data, along with previous studies, suggest an evolutionarily conserved relationship between SoxC and Twist proteins. Furthermore, our work identified new interactions in the gene regulatory network (GRN) underlying *C. elegans* postembryonic development and adds to the general understanding of the structure-function relationship of SoxC proteins.

## Author summary

SoxC transcription factors play important roles in metazoan development. Abnormal expression or function of SoxC factors has been linked to a variety of developmental disorders and cancers. It is therefore critical to understand the functions of SoxC proteins in vivo. *C. elegans* has a single SoxC transcription factor, SEM-2, which is known to regulate a fate decision between a proliferative progenitor cell vs. a terminally differentiated cell during postembryonic mesoderm development. In this study, we report new functions of SEM-2 in postembryonic mesoderm development via our studies of a partial loss-of-

**Data Availability Statement:** Strains and plasmids are available upon request. The authors affirm that all data necessary for confirming the conclusions

of the article are present within the article and its supporting information files.

**Funding:** Some strains were obtained from the C. elegans Genetics Center, which is funded by National Institutes of Health (NIH) Office of 27 Research Infrastructure Programs (P40 OD-010440). This work was supported by NIH R35 GM130351 to J.L.. M.B. was partially supported by the HHMI Gilliam Fellowship for Advanced Study (#GT13366) and the Cornell IMSD, which is funded by NIH R25 GM125597. A.L. was partially funded by the Cornell University College of Agriculture and Life Sciences Charitable Trust Grant and Morley Student Research Grant. L.P. and A.N.M. were Hunter R. Rawlings III Presidential Research Scholars at Cornell University. The funders had no role in study design, data collection and analysis, decision to publish, or preparation of the manuscript.

**Competing interests:** The authors have declared that no competing interests exist.

function allele of *sem-2*. Our work uncovers new regulatory relationships between SEM-2/SoxC and the FoxF/C transcription factor LET-381, and between SEM-2/SoxC and the *C. elegans* Twist ortholog HLH-8. Our findings suggest that the SoxC-Twist axis, including the downstream targets of Twist, represents an evolutionarily conserved regulatory cassette important in metazoan development.

## Introduction

Metazoan development is characterized by the specification and diversification of multipotent cells, as well as the proper organization of their differentiated descendants. These processes require well-orchestrated interplay between cell-intrinsic transcription factors and cell-cell signaling. The *C. elegans* postembryonic mesoderm, the M lineage, offers a unique model system to dissect the underlying regulatory logic of cell fate specification and diversification. The M lineage is derived from a single multipotent precursor cell, the M mesoblast [1]. During hermaphrodite postembryonic development, the M mesoblast cell undergoes stereotypical divisions to produce fourteen striated body wall muscles (BWMs), two non-muscle coelomocytes (CCs), and two multipotent sex myoblasts (SMs) that subsequently proliferate to produce sixteen sex muscles—four type I and four type II vulval muscles (vm1s and vm2s), as well as four type I and four type II uterine muscles (um1s and um2s)—that are required for egg laying (Fig 1A–1B).

Previous studies have identified multiple transcription factors and signaling components essential for the proper development of the M lineage [2]. In particular, LIN-12/Notch signaling is known to function upstream of the single *C. elegans* SoxC protein, SEM-2, to specify the SM fate in the ventral M lineage, while the zinc finger transcription factor SMA-9 antagonizes BMP signaling to specify the M lineage-derived CC (M-CC for short) fate in the dorsal side [3–6]. SMA-9 functions by activating the expression of the sole FoxF/C transcription factor in *C. elegans*, LET-381, in the M-CC mother cells. LET-381 then directly activates the expression of the Six homeodomain transcription factor, CEH-34, where LET-381 and CEH-34 function in a feedforward manner to directly activate the expression of genes required for CC differentiation and function [7,8]. At the same time, SMA-9 and LET-381 are each known to repress the expression of *sem-2* in the dorsal side of the M lineage to prevent it from specifying the SM fate [6].

In addition to the factors described above that are important for proper fate specification in the M lineage, the sole *C. elegans* Twist ortholog, HLH-8, is known to function in proper patterning of the M lineage. HLH-8 is a basic helix-loop-helix (bHLH) transcription factor that is expressed in the undifferentiated cells of the M lineage through regulatory elements located in the promoter, and it is expressed in the vulval muscles by autoregulation through E boxes located in its first intron [9,10]. HLH-8 is known to have multiple functions during M lineage development: proper cleavage orientation of M lineage cells, proper proliferation of the SMs, and proper differentiation and function of the vulval muscles [11,12].

In this study, we provide new insight into the relationships between various factors important in M lineage development and the sole SoxC transcription factor SEM-2. SoxC proteins are Sry-related HMG box (Sox)-containing transcription factors that are known to play critical roles in multiple developmental processes [13]. There are three highly conserved SoxC proteins in vertebrates, Sox4, Sox11, and Sox12. Abnormal expression or function of SoxC factors has been linked to a variety of developmental disorders and cancers [13–18]. In particular, mutations in Sox4 and Sox11, most of them being point mutations in the HMG box, are

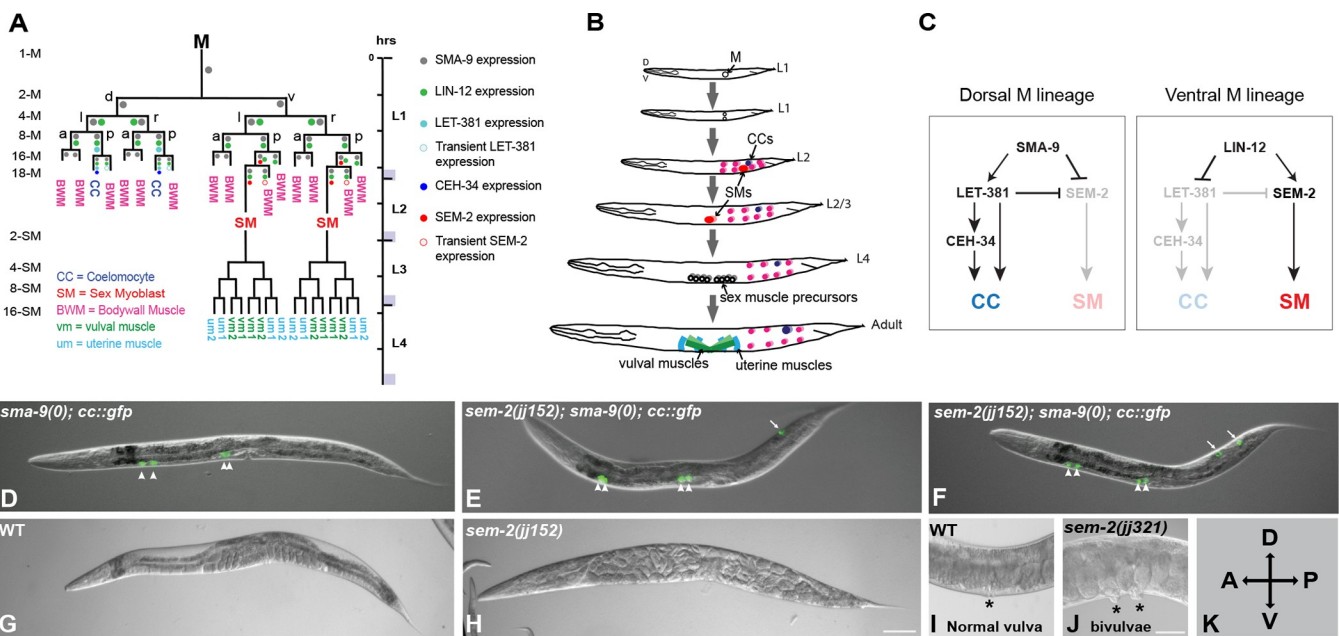

**Fig 1. The *sem-2(P158S)* mutants exhibit multiple defects during postembryonic development.** A) Diagram of the *C. elegans* hermaphrodite postembryonic mesoderm, M lineage, showing all differentiated cell types that arise from the M mesoblast cell, as well as the expression patterns of key factors critical for CC and SM fate specification in early M lineage development, based on previous publications [4,6–8,24,59,60]. a, anterior; p, posterior; d, dorsal; v, ventral; l, left; r, right. B) Schematic of developing *C. elegans* hermaphrodites showing the locations of M lineage cells. C) Model showing how SEM-2 is involved in SM vs. CC fate specification based on previous publications [4,6–8,24,59,60]. D) An L4 *sma-9(0)* animal that has four embryonic coelomocytes (arrowheads) labelled by *CC::gfp*. E–F) Two L4 *sem-2(jj152[P158S]); sma-9(0)* animals that have four embryonic CCs (arrowheads) and one (E) or two (F) M-CCs (arrows). G) A wild-type gravid adult hermaphrodite. H) A *sem-2(jj152[P158S])* gravid adult hermaphrodite with a uterus filled with late-stage embryos, showing the Egl phenotype. I) The vulva (asterisk) region of a wild-type gravid adult hermaphrodite. J) A *sem-2(jj321[P158S])* gravid adult hermaphrodite with two vulvae (asterisks). J) Orientation of all animals shown in this and subsequent figures. Scale bars in D-H represent 100 μm, while scale bars in I and J represent 50 μm.

associated with a developmental disorder called Coffin-Siris syndrome (CSS) [19–21]. However, the underlying molecular mechanisms are not completely understood.

We identified a new allele of *sem-2*, *jj152*, which is a point mutation resulting in a single amino acid change in a highly conserved residue at the end of the DNA-binding domain of SEM-2. We present genetic evidence showing that this single amino acid change results in a partial loss of SEM-2 function. By analyzing the M lineage phenotypes of *jj152* mutants, we uncovered an unexpected role of SEM-2 in the dorsal M lineage, where SEM-2 functions antagonistically at the level of expression and function with LET-381 in CC specification. We also found that SEM-2 regulates the expression of *hlh-8*, possibly directly, in the SMs, and that SEM-2 is essential for the proliferation and the diversification of the SMs. Our work uncovered new interactions in the gene regulatory network underlying *C. elegans* postembryonic development, some of which are likely conserved in other species, and we add to the general understanding of the functions and structure-function relationship of SoxC proteins.

## Materials and methods

### *C. elegans* strains and transgenic lines

*C. elegans* strains used in this study were maintained at 20˚C under normal culture conditions [22]. Analyses of *hlh-8* reporters in *sem-2(jj152)*, *sem-2(jj321)*, *sem-2(jj382 jj417)* and *sem-2 (jj476)* were performed at 25˚C unless specifically noted. All strains are listed in S1 Table.

## Microscopy

Epifluorescence and differential interference contrast (DIC) microscopy was conducted on a Leica DMRA2 compound microscope equipped with a Hamamatsu Orca-ER camera using the iVision software (Biovision Technology). Subsequent image analysis was performed using Fiji [23]. For comparison of fluorescence intensities in different genetic backgrounds, images were collected at the same magnification and exposure.

## Statistical analysis

Statistical significance was determined by performing unpaired two-tailed Student's *t*-tests or ANOVA with Dunnett's test using Prism10 (https://www.graphpad.com/features). Raw data for all statistical analysis are available in S1 Data.

## Isolation and mapping of *sem-2(jj152)*

*sem-2(jj152)* was isolated in a large-scale *sma-9(cc604)* Susm screen (suppression of the *sma-9 (cc604)* M lineage defect) for the restoration of M-derived CCs in *sma-9(cc604)* mutants [5]. The *sma-9(cc604)* allele is a nonsense mutation that is considered a genetic null [24] and referred to as *sma-9(0)*. *jj152* was mapped to chromosome I using the whole genome sequencing approach described in Liu et al. [5]. Because *jj152* animals are egg-laying defective (Egl), we performed complementation tests between *jj152* and each of the two previously studied *sem-2* alleles, *n1343* and *ok2422* [6]. All *jj152/n1343* trans-heterozygotes are Egl, but they did not exhibit the Susm phenotype. To perform a complementation test between *jj152* and *ok2422*, *hT2[bli-4(e937) let-?(q782) qIs48(myo-2p::gfp; pes-10p::gfp; ges-1p::gfp)] (I;III)/jj152 (I); jjIs3900(hlh-8p::NLS::mCherry::lacZ + myo-2p::mCherry) (IV)/+; sma-9(0) (X)* males were mated with *hT2[bli-4(e937) let-?(q782) qIs48(myo-2p::gfp; pes-10p::gfp; ges-1p::gfp)] (I;III)/sem-2(ok2422) (I); sma-9(0)* (X) hermaphrodites. *Red*, *non-green jjIs3900(hlh-8p::NLS::mCherry:: lacZ + myo-2p::mCherry) (IV)/+; jj152/sem-2(ok2422) (I); sma-9(0) (X)* cross progeny were scored for CC number. This cross scheme ensures the unambiguous identification of *jj152/ sem-2(ok2422); sma-9(0)* animals from the cross because 50% of the cross progeny will be *jjIs3900(hlh-8p::NLS::mCherry::lacZ + myo-2p::mCherry) (IV)/+*, which will have a red pharynx. Among these red cross progeny, the non-green animals will have the *jj152/sem-2(ok2422) (I)* genotype for the *sem-2* locus due to the lack of the *hT2[bli-4(e937) let-?(q782) qIs48(myo-2p::gfp; pes-10p::gfp; ges-1p::gfp)] (I;III)* balancer chromosome. Most *jjIs3900/+; jj152/sem-2 (ok2422); sma-9(0)* cross progeny were dead as embryos. Among those that survived, all (8/8) were Egl and exhibited the Susm phenotype.

## Suppression of the *sma-9(0)* M lineage defect (Susm) assay

For the Susm assay, animals containing a *CC::gfp* marker (see S1 Table) were grown at 20˚C to the young adult stage. The number of animals with four, five, or six CCs were tallied for each genotype. Statistical significance was calculated by performing unpaired two-tailed Student's *t*-tests.

## Assays for brood size and embryonic lethality

Brood size and embryonic lethality were scored under the Nikon SMZ1500 fluorescence stereo microscope. Single L4 animals were placed onto NGM plates and allowed to produce progeny at 20˚C. To score the brood size of wild-type N2 animals, the parent was transferred to a new plate every 24 hours for a total of 3 days, and the eggs and newly hatched L1 larvae on each plate were counted. To score the brood size of the Egl strains, the total number of eggs present

in the bagged parent and newly hatched larvae on the plate were counted 24–48 hours post-plating. To assess embryonic lethality, the progeny of all strains were allowed to grow to adulthood, and the total number of adults was noted and compared with the corresponding brood size of each parent. The average brood size and embryonic lethality were calculated for each strain. Statistical significance between the mutant strains and wild-type N2 worms was calculated by performing unpaired two-tailed Student's *t*-tests.

## Plasmids

All plasmids generated in this study and their purposes are described in S2 Table. pMDB28 (the homologous repair template used to generate endogenous *gfp*::*2xflag*::*sem-2*) was generated by performing Gibson Assembly with *sem-2* genomic sequences amplified from N2 worms, the sequence of *gfp* amplified from pDD282 [25], a gBlock containing *2xflag* purchased from Integrated DNA technologies (IDT), and a pBSII/SK+ vector digested with BamHI and HindIII. pMDB37 (*the homologous repair template for generating hlh-8(jj422[hlh-8p::hlh-8::sl2::nls::gfp::nls::hlh-8 3'UTR]))* was generated by performing Gibson Assembly with *hlh-8* genomic sequences amplified from N2 worms, a gBlock with the *gpd-2* intergenic region and nuclear localization sequence from CEOPX036 [26], the sequence of *gfp* amplified from pDD282 [25], another nuclear localization signal introduced by primer MDB-87, and a pBSII/SK+ vector digested with BamHI and HindIII.

pAYL11, pAYL21–pAYL25, and pAYL31–pAYL35 (*hlh-8* promoter deletion constructs) were generated using pJKL502 in the 1999 Fire Lab Vector Kit as the original template. pJKL502 contains a 1.3kbp fragment (-1.3kbp to -1bp) of the *hlh-8* promoter driving *gfp* expression. As a starting point, deletions were made at either the 5' or 3' end of a 517bp fragment (-517bp to -1bp) of the *hlh-8* promoter, which was previously shown to be sufficient to drive M lineage expression [9]. These fragments contained HindIII and XbaI restriction sites at their ends and were used to replace the promoter region in pJKL502. Next, the 50bp internal deletion constructs, pAYL21–pAYL25, were generated by ligating the promoter elements from a 5' deletion construct and a 3' deletion construct, thus resulting in a set of constructs each missing a contiguous 50 bp sequence of the *hlh-8* 517bp promoter. Constructs that contain 20bp deletions in the -300bp to -200bp region, pAYL31–pAYL35, were generated by a two-step bridging/fusion PCR scheme where PCR fragments containing an overlapping 20bp region were used as templates for a second round of PCR to bridge the two fragments together. The resulting fragments (flanked by HindIII and XbaI sites) were subcloned to replace the promoter region in pJKL502 in the same way as described for the 5' and 3' deletion constructs. All plasmids were confirmed by Sanger sequencing.

## CRISPR

CRISPR experiments were conducted either by using plasmids expressing regular Cas9 (pDD162) [25] or the VQR variant of Cas9 (pRB1080) [27] and plasmids expressing sgRNAs in the pRB1017 backbone [28], or by injecting ribonuclear RNP complexes with Cas9 protein, tracer RNA (from IDT), and sgRNA as described in Beacham et al. [29] (sequences listed in S3 Table). For large insertions, plasmid repair templates (S2 Table) were used, while ssDNA oligos (S3 Table) were used as repair templates to introduce point mutations. For injections, pRF4(*rol-6(d)*) [30] was used as a co-injection marker. Injected animals were singled onto NGM plates seeded with OP50 bacteria. Plates that gave the most roller progeny were selected for screening by PCR. Final CRISPR edits were confirmed by Sanger sequencing.

## RNAi

The plasmids for *let-381(RNAi)* and *ceh-34(RNAi)* were obtained from the Ahringer RNAi library [31] and confirmed by sequencing. RNAi was conducted by following the protocol of Amin et al. [7]. Synchronized L1 animals of various genotypes were plated on HT115(DE3) bacteria expressing dsRNA against the gene of interest, allowed to grow at 25°C, and scored for M lineage phenotypes 12–48 hours after plating. Bacteria carrying the L4440 empty vector was used as a negative control.

## Results

### A *sem-2* allele, *jj152*, suppresses the *sma-9(0)* M lineage phenotype in coelomocyte specification

In a *sma-9(0)* suppressor screen to identify new factors involved in M lineage development [5], we identified a mutation, *jj152*, on chromosome I. *jj152* animals showed partial suppression of the *sma-9(0)* M lineage (Susm) phenotype: 41.2% of *jj152; sma-9(0)* animals (N = 767) have 1–2 M-CCs, instead of zero M-CCs in *sma-9(0)* single mutant (Fig 1D–1F, Table 1). *jj152* animals are also 100% egg-laying defective (Egl), have a smaller brood size than wild-type (WT), display ~25% embryonic lethality and exhibit a 6.9% bivulva (Biv) phenotype (N = 360) (Fig 1G–1J, Table 2). Unlike *sma-9(0)* mutants, which have a small body size and lack both M-CCs, *jj152* single mutants are not small, and all *jj152* mutants have 1–2 M-CCs (Table 1).

Several lines of evidence suggest that *jj152* is an allele of *sem-2*: 1) whole genome sequencing revealed that *jj152* maps to chromosome I and contains a cytosine (C) to thymine (T) nucleotide change, which results in a proline (P) to serine (S) residue change in amino acid 158 (P158S) of SEM-2 (Fig 2A–2B); 2) *jj152* failed to complement the *sem-2(ok2422)* null allele in both the Egl and the Susm phenotypes (Tables 1 and 2); 3) a transgene carrying a fosmid containing the wild-type *sem-2* genomic sequences (*jjIs1647[sem-2(+)]*, [6]) rescued both the Susm and the Egl phenotypes of *jj152* mutants (Tables 1 and 2); and 4) CRISPR-engineered *sem-2(P158S)* mutations in the wild-type background, *sem-2(jj320)* and *sem-2(jj321)*, recapitulated the mutant phenotypes exhibited by *jj152* animals (Tables 1 and 2). Given that *jj152*,

**Table 1. The *sem-2(P158S)* mutation suppresses the *sma-9(cc604)* missing M-derived coelomocyte (CC) phenotype.**

| Genotype | % 1–2 M-CCs[a] | % Egl | No. of animals |
|---|---|---|---|
| *sem-2(jj152)* | 100 | 100 | 845 |
| *sma-9(cc604)* | 1.4 | - | 1359[b] |
| *sem-2(n1343); sma-9(cc604)* | 0.59% | 100 | 1672 |
| *sem-2(jj152); sma-9(cc604)* | 41.2*** | 100 | 767 |
| *sem-2(jj152); sma-9(cc604); jjIs1647[sem-2(+)]* | 0 [ND] | 0 | 329 |
| *sem-2(ok2422)/+; sma-9(cc604)* | 2.4[ND] | 0 | 205 |
| *sem-2(ok2422)/sem-2(jj152); sma-9(cc604)* | 100*** | 100 | 8[c] |

[a] CCs were scored using the *CC::gfp* described in S1 Table.

[b] Data from Liu et al. (2022) [5].

[c] Most *sem-2(ok2422)/sem-2(jj152); sma-9(cc604)* animals are embryonic lethal, with only few animals able to survive through larval development to become young adults, which is when we score the M-CC phenotype. Genotyping four of the eight animals confirmed that they are *jj152/ok2422*. Details of the complementation tests are described in Materials and Methods.

ND: no difference

***: *P*<0.001, when compared to *sma-9(cc604)* animals based on unpaired two-tailed Student's *t*-test.

**Table 2. The *sem-2(P158S)* mutation results in reduced brood size, increased embryonic lethality, and an egg-laying defective phenotype.**

| Genotype | % Egl (n) | Brood size (n) | % Emb (n) |
|---|---|---|---|
| WT | 0 (>100) | 303 ± 26.7 (9) | 0 (2582) |
| *sem-2(jj152)* | 100 (>100) | 32.4 ± 8.3 (11)*** | 17.2 ± 9.7 (314)*** |
| *sem-2(jj320)* | 100 (>100) | 41.9 ± 5.5 (10)*** | 20.97 ± 8.4 (419)*** |
| *sem-2(jj321)* | 100 (>100) | 37.3 ± 5.2 (10)*** | 30.31 ± 9.2 (373)*** |
| *sem-2(ok2422)* | N/A | N/A | 100***[a] |

Egl: egg-laying defective; Emb: embryonic lethal.

The average brood size and average percentage of Egl or Emb ± standard deviation are presented in this table. Details on scoring the brood size and embryonic lethality are described in Materials and Methods.

n, number of worms scored. For brood size, n reflects the number of parents singled and their brood sizes scored.

***: $P<0.001$ when compared to WT based on unpaired two-tailed Student's *t*-test. N/A: not applicable.

[a] data from Tian et al. 2011 [6].

*jj320*, and *jj321* mutants exhibit a weaker phenotype than *sem-2(ok2422)* null or trans-heterozygous mutants (Tables 1–2), or animals that have undergone postembryonic *sem-2(RNAi)* [6], we concluded that the SEM-2(P158S) mutant protein exhibits a partial loss of SEM-2 function.

## The single amino acid change in *sem-2(jj152)*, P158S, does not drastically affect the localization or level of SEM-2 protein

We were intrigued by the Susm phenotype of *sem-2(jj152[P158S])* because *sem-2(n1343)*, which has a Tc1 transposon insertion that disrupts *sem-2* expression in the SM lineage and exhibits a SM to BWM fate transformation phenotype, does not show any Susm phenotype (Table 1, [6]). We therefore decided to further explore the impact of the *sem-2(jj152[P158S])* point mutation on SEM-2 function, and the relationship between *sem-2* and *sma-9* in M-CC fate specification.

*sem-2* encodes the sole SoxC protein in *C. elegans*, whereas vertebrates have three SoxC proteins, Sox4, Sox11, and Sox12 [32]. SoxC proteins have a conserved DNA-binding domain, a serine-rich region, and a transactivation domain (Fig 2A). Amino acid P158 is located near the end of SEM-2's DNA-binding domain and is conserved in SoxC homologs from vertebrates (Fig 2A). Structural modeling based on the co-crystal structure of mouse Sox4 and its cognate DNA showed that P158 is located in a flexible linker region of the domain but does not directly touch DNA (Fig 2B, [33]). It is likely that the P158S mutation alters the conformation of SEM-2, thus affecting its DNA-binding affinity or its affinity to other factors that function together with SEM-2.

To begin to assess the functional consequences of the P158S mutation on SEM-2 protein, we tagged SEM-2 at the endogenous locus with GFP::2xFLAG using CRISPR. This endogenously tagged SEM-2 is functional as GFP::2xFLAG::SEM-2 animals are viable, fertile, and non-Egl. As shown in Fig 2C, GFP::2xFLAG::SEM-2 is nuclear localized in many different cell types, such as cells of the hypodermis, intestine, vulva and pharynx, similar to what we have previously reported using transgenic animals expressing a GFP-tagged SEM-2 in a fosmid backbone [6]. To determine if the P158S mutation affects SEM-2 expression, localization, or stability, we used CRISPR and introduced the P158S mutation into the endogenously tagged GFP::2xFLAG::SEM-2, and generated *sem-2(jj382 jj417[GFP::2xFLAG::SEM-2(P158S)]*. As shown in Fig 2C–2E, SEM-2(P158S) exhibits a similar pattern and level of expression and

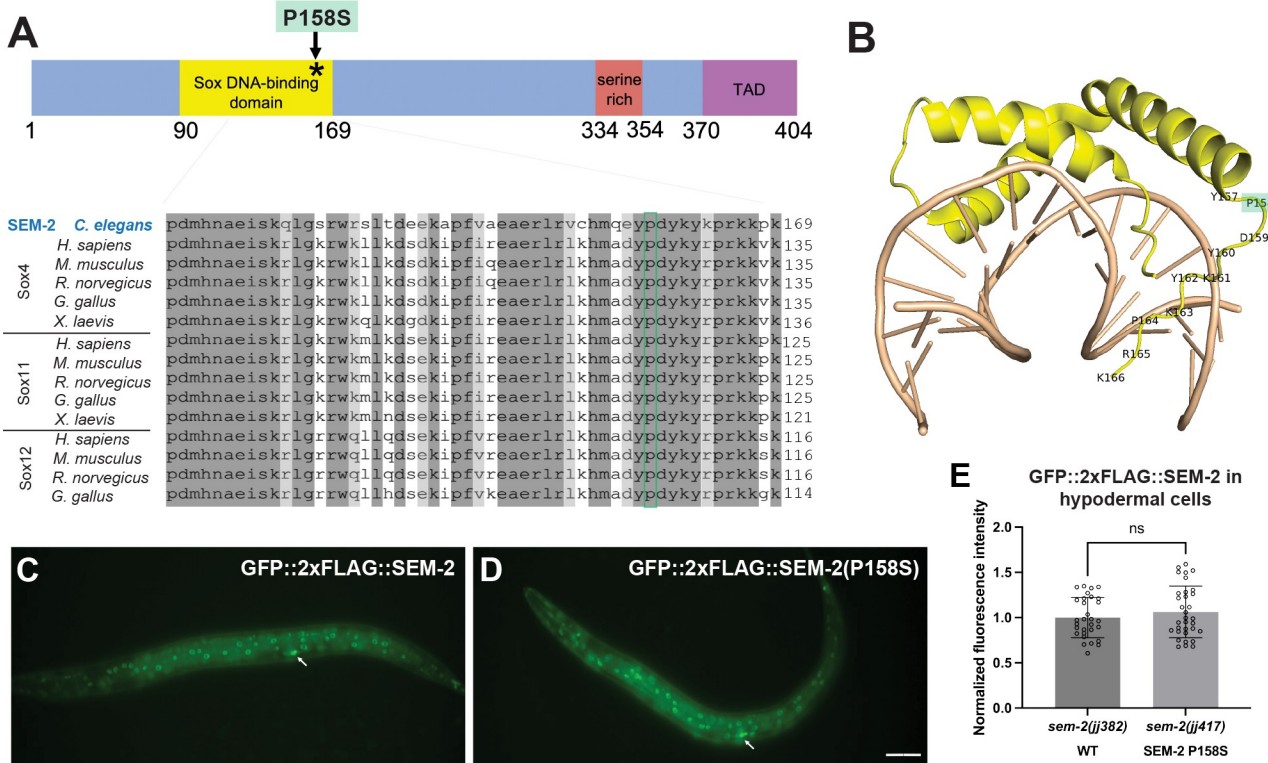

**Fig 2. The *sem-2(P158S)* mutation does not affect the expression, localization, or level of the SEM-2 protein.** A) [Top] Schematic of the SEM-2 protein with the DNA-binding domain in yellow, the serine-rich region in red, and the transactivation domain (TAD) in purple. The P158S mutation is indicated by an asterisk. [Bottom] Sequence alignment of part of the DNA-binding domain of SEM-2 with SoxC proteins in different vertebrate species. Residue P158 is conserved in all SoxC proteins shown and is highlighted by a green box. B) Structural model of the SEM-2 DNA-binding domain (yellow) with DNA (tan), based on the structure of the Mouse Sox4 DNA-binding domain-DNA complex (PDB code 3U2B [33]. Residue P158 is highlighted in green. C–D) Fluorescence images showing the expression patterns of the endogenously tagged GFP::2xFLAG::SEM-2 (C) and GFP::2xFLAG::SEM-2(P158S) (D) in hermaphrodites at the L3 stage. Arrows point to the migrating SMs in the focal plane. Scale bar represents 40 μm. E) Quantification of GFP::2xFLAG::SEM-2 fluorescence intensity in hypodermal nuclei of WT (*sem-2(jj382)*) and *sem-2(P158S)* (*sem-2(jj417)*) animals. Three nuclei per animal were measured. Each dot represents a nucleus. Data are normalized to WT. Statistical significance was calculated by performing unpaired two-tailed Student's *t*-tests. ns, not significant.

localization to the wild-type SEM-2 protein. Considering the structural information shown in Fig 2B, we concluded that the P158S mutation likely affects the activity of the SEM-2 protein. This prediction is consistent with our genetic evidence suggesting that SEM-2(P158S) is partially functional.

### *sem-2* exhibits dynamic expression patterns in the postembryonic M lineage

Having an endogenously tagged GFP::2xFLAG::SEM-2 allowed us to more accurately assess the expression pattern of *sem-2* in the M lineage. Consistent with our previous report based on transgenic animals expressing GFP::SEM-2 in a fosmid backbone [6], endogenous GFP::2xFLAG::SEM-2 expression is detectable in the SM mothers, the SMs, and throughout the SM lineage in all SM descendants before they terminally differentiate (Fig 3E–3K"). Surprisingly, strong nuclear GFP::2xFLAG::SEM-2 signal is detectable in the M mesoblast, and it remains detectable, but becomes progressively fainter, in all M lineage cells up to the 16-M stage (Fig 3A–3E). At the 16-M stage, GFP::2xFLAG::SEM-2 signal is significantly brighter in the SM mother cells (M.vlpa and M.vrpa) (Figs 3E–3E" and S1G). At the 18-M stage when the

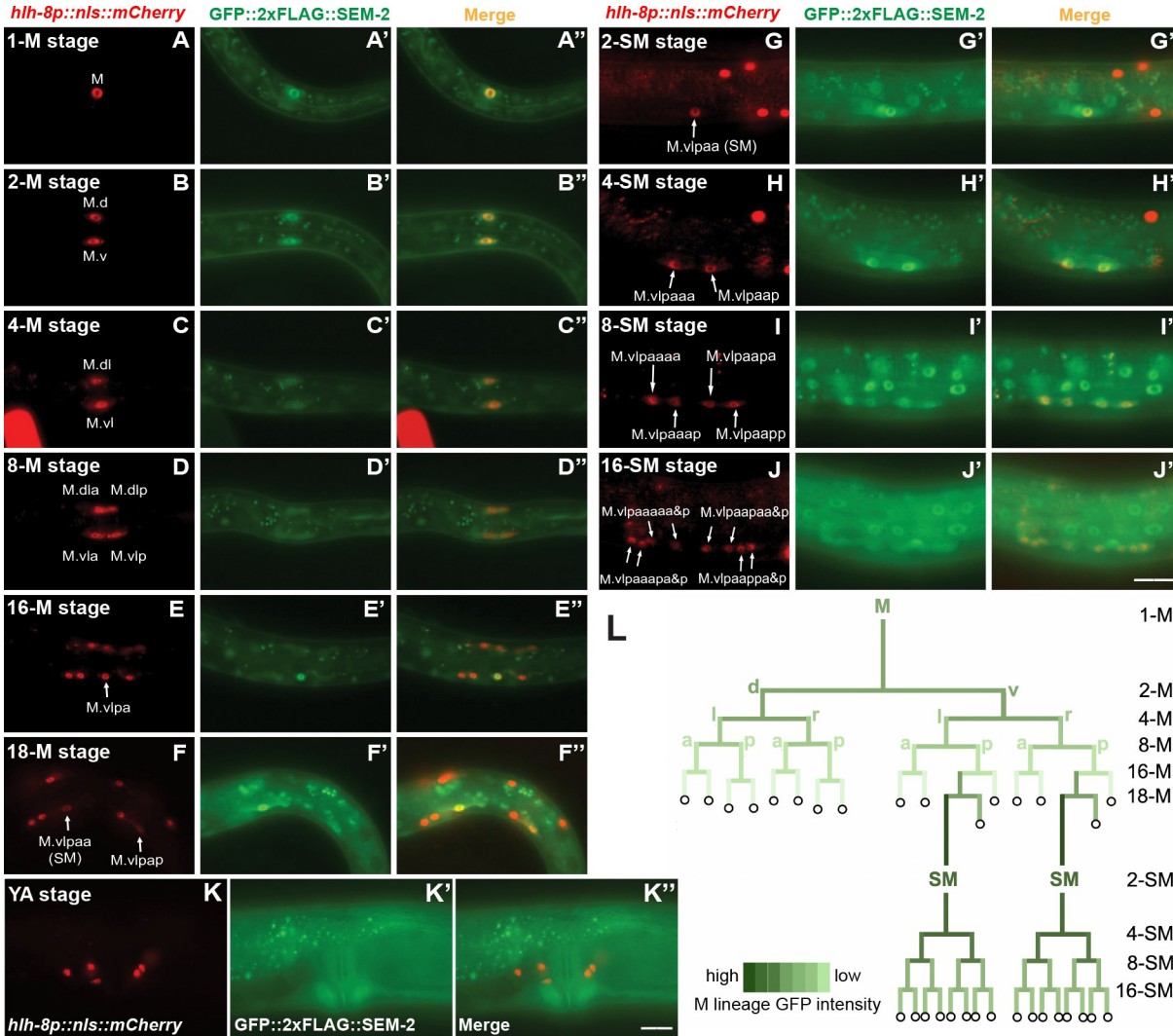

**Fig 3. Endogenously tagged GFP::2xFLAG::SEM-2 exhibits dynamic expression throughout the M lineage.** A–K") Fluorescence images showing the expression of GFP::2xFLAG::SEM-2 (A'–K') in M lineage cells of hermaphrodites labelled by the *hlh-8p::nls::mCherry* reporter (A–K) at different stages of M lineage development, with the corresponding merged images shown in A"–K". The SM sisters (M.vlpap and M.vrpap) migrate posteriorly after they are born and become BWMs [1]. Only the left side of an animal is shown in A–K". The other side is out of the focal plane. YA, young adult. Scale bar represents 20 μm for A–J" and 10 μm for K–K". L) Summary of GFP::2xFLAG::SEM-2 expression in the M lineage, based on quantification of nuclear GFP intensity shown in S1 Fig. Cells represented by the light green color have lower levels of GFP::2xFLAG::SEM-2 expression than cells represented by the dark green color. Circles with black outline represent terminally differentiated cells that do not express GFP::2xFLAG::SEM-2.

BWMs and CCs are born and become terminally differentiated, GFP::2xFLAG::SEM-2 is only detectable in the two SMs (M.vlpaa and M.vrpaa) and transiently in the SM sister cells (M. vlpap and M.vrpap) before they terminally differentiate into BWMs (Fig 3F–3F"). At all stages of M lineage development when GFP::2xFLAG::SEM-2 is detectable, there is GFP signal in both the nucleus and the cytoplasm. The intensity of both nuclear and cytoplasmic GFP signals exhibit a gradual decrease during the two proliferative phases of M lineage development (in the early M lineage and in the SM lineage) (S1 Fig). Moreover, the level of GFP::2xFLAG:: SEM-2 fluorescence intensity in the SM cells is significantly higher than in the M cell (S1 Fig). The expression and localization pattern of GFP::2xFLAG::SEM-2 in the M lineage is quantified and summarized in Figs 3L and S1.

To determine if the P158S mutation affects the expression and/or localization of SEM-2 specifically in the M lineage, we compared the expression and localization pattern of wild-type GFP::2xFLAG::SEM-2 with mutant GFP::2xFLAG::SEM-2(P158S) (S2 Fig). At the 1-M stage, we detected a slight increase in nuclear GFP intensity as well as nuclear to cytoplasmic ratio in *sem-2(P158S)* mutants compared to wild-type animals (S2C and S2E Fig). However, there is no difference in SEM-2 level or localization in the SMs when comparing wild-type and *sem-2 (P158S)* mutant animals (S2F–S2L Fig).

## SMA-9 specifically represses the dorsal M lineage expression of GFP::2xFLAG::SEM-2 at the stage of CC and SM fate specification

We have previously shown that SMA-9 represses the expression of SEM-2 in the dorsal M lineage at the 18-M stage when the SMs are born [6]. Given the unexpected Susm phenotype of *sem-2(jj152[P158S])* and the unexpected expression of endogenous GFP::2xFLAG::SEM-2 in the early M lineage prior to the birth of SMs, we sought to determine if the expression level or pattern of GFP::2xFLAG::SEM-2 changes in the M lineage of *sma-9(0)* mutants. Consistent with our previous report, we observed expression of endogenous GFP::2xFLAG::SEM-2 in the SM-like cells in the dorsal M lineage of *sma-9(0)* animals (Fig 4B–4B"'). However, we did not detect any difference in the expression level or pattern of GFP::2xFLAG::SEM-2 between WT and *sma-9(0)* animals from the 1-M to the 8-M stage (S3 Fig). These results suggest that SMA-9 functions to repress the dorsal M lineage expression of *sem-2* only at the stage of CC and SM fate specification.

## SEM-2 antagonizes the function of LET-381 in specifying M-derived CCs by repressing *let-381* expression

As described above, when the fully functional SEM-2 is ectopically expressed in the dorsal M lineage of *sma-9(0)* single mutants, no M-CCs are produced. However, around 40% of *sem-2 (jj152[P158S]); sma-9(0)* double mutants have 1–2 M-CCs. The forkhead transcription factor LET-381/FoxF/C is known to function downstream of SMA-9 and upstream of the Six homeodomain transcription factor CEH-34, where LET-381 and CEH-34 function in a feedforward manner to specify the M-CC fate [7,8]. We therefore hypothesized that the Susm phenotype of *sem-2(jj152[P158S])* might be because the partially functional SEM-2(P158S) protein, when ectopically expressed in the dorsal side of the M lineage at the 16-M stage, cannot fully inhibit either the expression or the function of *let-381*.

To test the above hypothesis, we examined the expression pattern of an endogenously tagged mNG::LET-381 [34] in WT, *sma-9(0)*, and *sem-2(jj476[P158S]); sma-9(0)* animals. Since *sem-2* and *let-381* are located close to each other on Chromosome I (*sem-2* at -0.27 while *let-381* at +1.02), we used CRISPR to introduce the *P158S* mutation into the *let-381(dev205 [mNG::LET-381])* strain and generated the *sem-2(jj476[P158S]) let-381(dev205[mNG::LET-381])* strain. We then conducted genetic crosses and generated the *sem-2(jj476[P158S]) let-381 (dev205[mNG::LET-381]); sma-9(0)* strain. Similar to the previously reported pattern of expression for the LET-381::GFP transgene [8], endogenous mNG::LET-381 is expressed in the dorsal M-CC mothers (M.dlp and M.drp) and the CCs (M.dlpa and M.drpa) in WT animals (100%, N = 40) (Fig 4E–4E"') and in *sem-2(jj476[P158S])* single mutants (98%, N = 50, Fig 4F–4F"'), while this M lineage expression disappears in *sma-9(0)* mutants (98.3%, N = 59) (Fig 4G–4G"'). Instead, 65.9% of *sem-2(jj476[P158S]) let-381(dev205 [mNG::LET-381]); sma-9 (0)* mutants examined (N = 41) showed expression of mNG::LET-381 in M.dlpa and/or M. drpa (Fig 4H–4H"'), cells that are normally fated to become CCs. Consistent with this finding, the M-CCs formed in *sem-2(jj152[P158S]); sma-9(0)* double mutants require LET-381. As

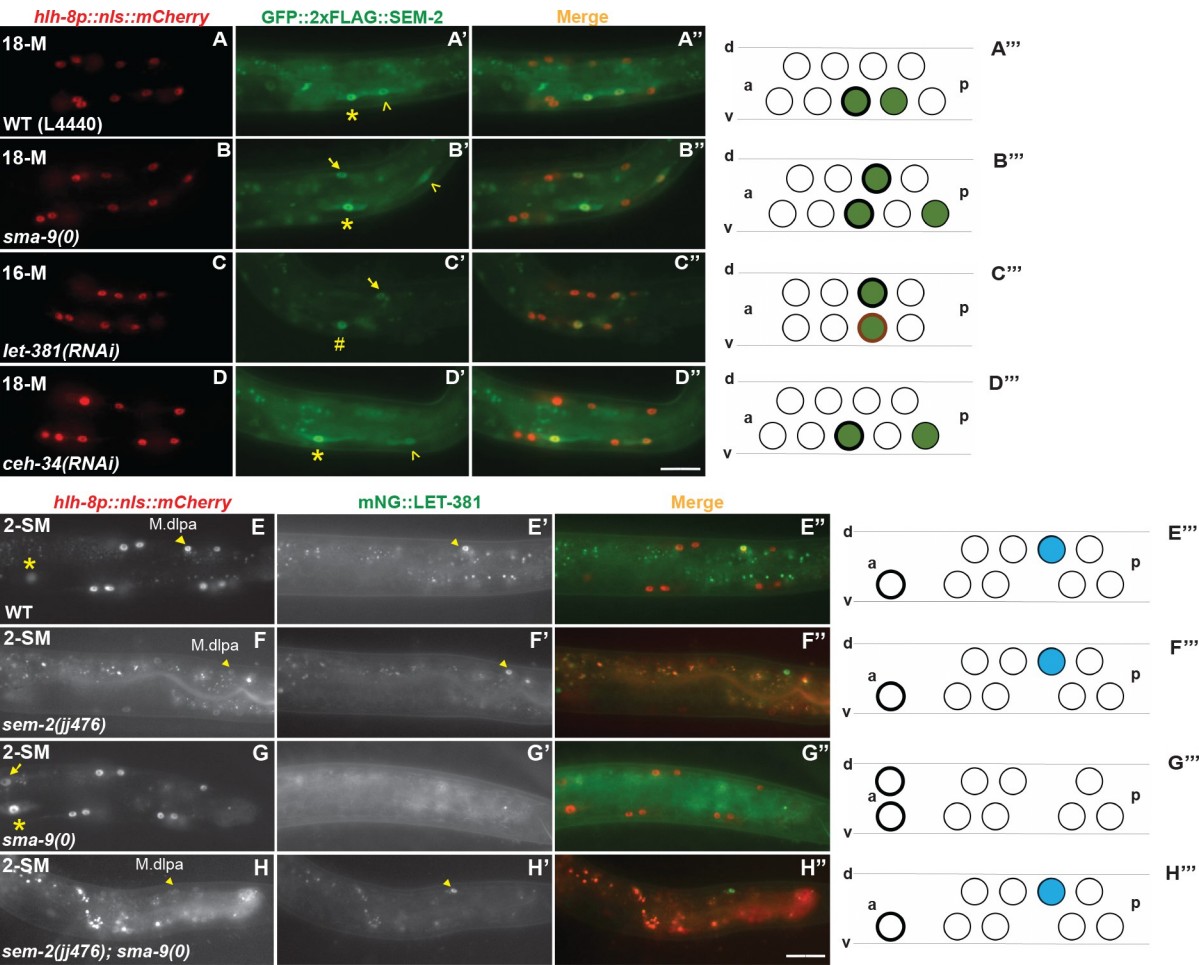

**Fig 4. SMA-9 and LET-381 repress *sem-2* expression in the dorsal M lineage, while ectopically expressed SEM-2 represses *let-381* expression in the dorsal M lineage.** A–D") Fluorescence images showing GFP::2xFLAG::SEM-2 (A'–D') in M lineage cells labelled by the *hlh-8p::nls::mCherry* reporter (A–D) at different stages of M lineage development in WT (A–A"), *sma-9(0)* (B–B"), *let-381(RNAi)* (C–C"), and *ceh-34(RNAi)* (D–D") hermaphrodites. A"–D" are corresponding merged images. A'''–D''') Schematic representation of the fluorescence images. Green circles represent cells expressing GFP::2xFLAG::SEM-2. Green circles with thick, black outlines are SMs. Green circles with thin, black outlines represent SM sisters fated to become BWMs. The cell with thick, burnt orange outline is an SM mother (C'''). E–H") Fluorescence images showing mNG::LET-381 (E'–H') in M lineage cells labelled by the *hlh-8p::nls::mCherry* reporter (E–H) at different stages of M lineage development in WT (E–E"), *sem-2(jj476[P158S])* (F–F"), *sma-9(0)* (G–G"), and *sem-2(jj476[P158S]); sma-9(0)* (H–H") hermaphrodites. (E"–H") are corresponding merged images. E'''–H''') Schematic representation of the fluorescence images. Blue circles represent CC cells expressing mNG::LET-381. Circles with thick, black outlines are SMs that have migrated to the future vulva region. Yellow asterisks label the SM cells (M.vlpaa); yellow unshaded arrowheads label the SM sister cells (M.vlpap), which are BWMs; yellow number sign labels the SM mother cell (M.vlpa); yellow arrows label the SM-like cells born in the dorsal side of the M lineage in *sma-9(0)* (B', G) or *let-381(RNAi)* (C') hermaphrodites; yellow, shaded arrowheads label the CC cells (M.dlpa) in WT (E–E'), *sem-2(jj476[P158S])* (F–F'), and *sem-2(jj476[P158S]); sma-9(0)* (H–H') hermaphrodites. The fluorescence intensity of the *hlh-8p::nls::mCherry* reporter is significantly reduced in animals carrying the *sem-2(jj476[SEM-2(P158S)]* mutation and was adjusted individually for panels F and H for cell identification. The intensity of the *hlh-8p::nls::mCherry* reporter is hereafter not comparable, unless specifically noted. Only the left side of an animal is shown in this figure, while the other side is out of the focal plane. Scale bars represent 20 μm.

shown in Table 3, while 24.6% of *sem-2(jj152[P158S]); sma-9(0)* animals on control RNAi with the empty vector L4440 (N = 240) had 1–2 M-CCs, only 3.1% of *sem-2(jj152[P158S]) let-381 (RNAi); sma-9(0)* animals (N = 291) had M-CCs. Taken together, the above findings demonstrate that SEM-2 antagonizes the function of LET-381 in specifying M-derived CCs by repressing *let-381* expression.

**Table 3. LET-381 is required for the formation of M-derived CCs in *sem-2(jj152); sma-9(cc604)* double mutants.**

| Genotype | RNAi[a] | % with no M-CCs[b] | % with 1–2 M-CCs | No. of animals |
|---|---|---|---|---|
| WT | L4440 | 7.6 | 92.2 | 437[c] |
| WT | *let-381* | 93.4 | 6.6 | 412 |
| *sem-2(jj152)* | L4440 | 0.7 | 99.3 | 431 |
| *sem-2(jj152)* | *let-381* | 92.1 | 7.9 | 350 |
| *sem-2(jj152); sma-9(cc604)* | L4440 | 75.4 | 24.6 | 240 |
| *sem-2(jj152); sma-9(cc604)* | *let-381* | 96.9 | 3.1 | 291 |

[a] Postembryonic RNAi as described in Materials and Methods.

[b] CCs were scored using the *CC::gfp* described in S1 Table.

[c] One of the 437 animals scored had 3 M-CCs.

## LET-381, but not CEH-34, represses the expression of SEM-2 in dorsal M lineage cells fated to become CCs

We have previously shown that in *let-381(RNAi)* animals, CCs are transformed to SM-like cells in the dorsal side of the M lineage [8], and that LET-381 represses the expression of the GFP::SEM-2 transgene in the dorsal M lineage cells fated to become M-CCs [6]. We found that this result holds true for the endogenously tagged GFP::2xFLAG::SEM-2, as *let-381(RNAi)* animals had GFP::2xFLAG::SEM-2 ectopically expressed in the dorsal M lineage cells that are normally fated to become M-CCs (Fig 4C–4C'").

Because LET-381 and CEH-34 function in a feedforward manner to specify M-CCs, we also examined the expression pattern of GFP::2xFLAG::SEM-2 in *ceh-34(RNAi)* animals. As shown in Fig 4D–4D'", *ceh-34(RNAi)* animals did not exhibit any ectopic expression of GFP::2xFLAG::SEM-2 in the dorsal M lineage. This is consistent with CEH-34 functioning downstream of LET-381 for specifying M-CCs [7,8]. Thus, LET-381, but not CEH-34, functions to repress *sem-2* expression in cells fated to become M-CCs.

## SEM-2 regulates the expression of *hlh-8* in the M lineage

During our analysis of the mechanistic basis of the Susm phenotype of *sem-2(P158S)* mutants, we noticed that *sem-2(P158S)* mutants exhibit significantly reduced levels of expression of the transgenic *hlh-8* transcriptional reporter *jjIs3900[hlh-8p::nls::mCherry]* (Figs 5 and S4). In wild-type animals, *hlh-8p::nls::mCherry* is expressed in the M mesoblast and all undifferentiated cells in the M lineage from the 1-M to 16-SM stage. The *sem-2(jj321[P158S])* mutants display reduced expression of this reporter at all stages of M lineage development (100%, N = 234) (Figs 5A–5F and S4A–S4C). Similar reduction of *hlh-8p::nls::mCherry* expression was observed in the M lineage of *sem-2(jj152[P158S])* mutants (100%, N = 643). Moreover, the expression of another *hlh-8* transcriptional reporter *ayIs6[hlh-8p::gfp]* [9] is also reduced in *sem-2(jj152[P158S])* mutants (100%, N = 599) (S4D–S4G Fig). Thus, the expression of two independent, transgenic, *hlh-8* transcriptional reporters is significantly reduced in *sem-2(jj152 [P158S])* mutants.

To determine if the reduced expression of *hlh-8* in *sem-2(P158S)* mutants is specific to the *hlh-8* gene in the M lineage, we examined the expression of *mls-2* in *sem-2(jj321[P158S])* mutants using an endogenously tagged mNG::MLS-2 [35]. MLS-2 is a NK homeodomain protein that regulates patterning, cell fate specification, and proliferation in the early M lineage [36]. *mls-2* is expressed starting at the 1-M stage in the M lineage [36]. We found no change in expression level or pattern of mNG::MLS-2 in *sem-2(jj321[P158S])* mutants compared to WT

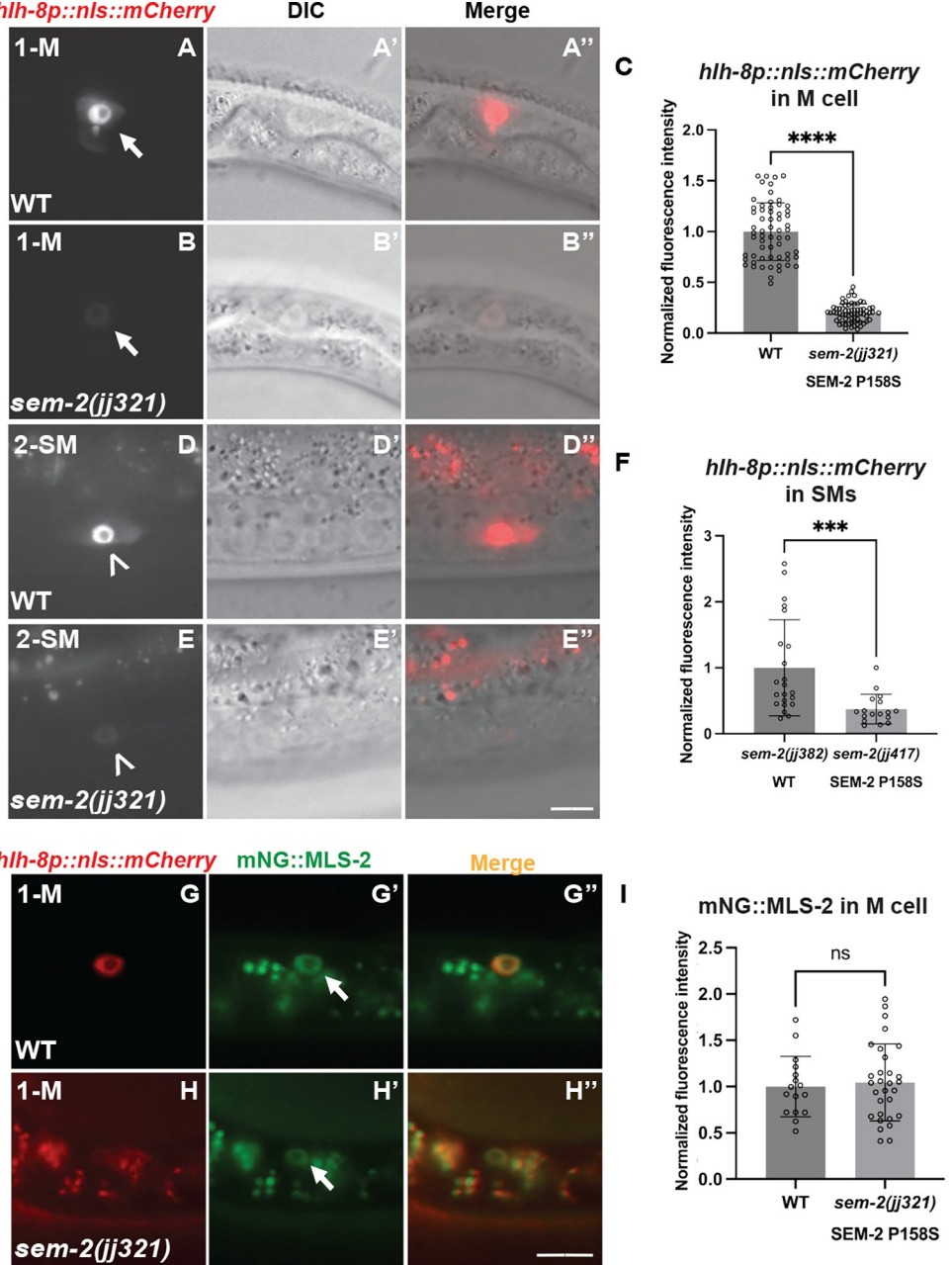

**Fig 5. SEM-2 regulates *hlh-8* expression in the M lineage.** A–C) Fluorescence images (A–B) of wild-type (A–A") and *sem-2(jj321[P158S])* (B–B") L1 animals showing expression of *hlh-8p::nls::mCherry* in the M mesoblast cell. A'–B' and A"-B" are corresponding DIC and merged images, respectively. C) Quantification of *hlh-8p::nls::mCherry* expression level in the M mesoblast of wild-type and *sem-2(jj321[P158S])* L1 animals. Each dot represents a nucleus. Data are normalized to WT. D–F) Fluorescence images (D–E) of wild-type (D–D") and *sem-2(jj321[P158S])* (E–E") L3 animals showing expression of *hlh-8p::nls::mCherry* in SMs. D'–E' and D"–E" are corresponding DIC and merged images, respectively. F) Quantification of *hlh-8p::nls::mCherry* expression in the SMs of wild-type and *sem-2(jj321[P158S])* L3 animals. Each dot represents a nucleus. Data are normalized to WT. For panels A–E, fluorescence images of the same staged animals were captured at the same exposure and magnification. G–I) Fluorescence images (G'–H') of wild-type (G–G") and *sem-2(jj321[P158S])* (H–H") L1 animals showing expression of mNG::MLS-2 in the M mesoblast labelled by the *hlh-8p::nls::mCherry* reporter (G–H). Arrows point to the M mesoblast cell, while arrowheads point to the SM cell. There is an increase in background signal in panels H and H" because a longer exposure time was used due to the reduced expression level of *hlh-8p::nls::mCherry* in *sem-2(P158S)* mutants. I) Quantification of mNG::MLS-2 in the M mesoblast cell of wild-type and *sem-2(jj321[P158S])* animals. Each dot represents a nucleus. Data are normalized to WT. Statistical significance was calculated by performing unpaired two-tailed Student's *t*-tests. **** *P*<0.0001, *** *P*<0.001, ns, not significant. Scale bars represent 10 μm.

(Fig 5G–5I). These results suggest that SEM-2 specifically regulates the expression of *hlh-8* in the M lineage.

To determine if the change in *hlh-8* expression was specific to the *sem-2(P158S)* allele, we introduced *jjIs3900[hlh-8p::nls::mCherry]* into *sem-2(ok2422)* null mutants. The majority of *sem-2(ok2422)* null mutants produced by heterozygous *sem-2(ok2422)/hT2[qIs48]* mothers die as threefold embryos, while a few embryos can hatch but die as L1 larvae [6]. We found that 83.3% *sem-2(ok2422)* embryos (N = 12) express *hlh-8p::nls::mCherry* in the M mesoblast cell at similar levels as *sem-2(ok2422)/hT2[qIs48]* control embryos (92.9%, N = 14) (S4H–S4J Fig). However, none of the *sem-2(ok2422)* L1 animals (100%, N = 14) display any detectable *hlh-8p::nls::mCherry* expression, unlike the *sem-2(ok2422)/hT2[qIs48]* control L1s where most of them show bright *hlh-8p::nls::mCherry* expression in the M mesoblast cell (98.1%, N = 52) (S4K–S4L''' Fig). One possible reason why there is more robust expression of *hlh-8p::nls::mCherry* in *sem-2(ok2422)* null embryos than in *sem-2(jj321[P158S])* embryos is maternal contribution of SEM-2 in the *sem-2(ok2422)* null embryos by *sem-2(ok2422)/hT2[qIs48]* parents (S4A–S4C and S4H–S4J Fig). Further, these results suggest that SEM-2 is required for maintaining stable *hlh-8* expression in the M lineage.

## SEM-2 is required for endogenous *hlh-8* expression in the M lineage

To determine if SEM-2 is required for endogenous *hlh-8* expression in the M lineage, we generated an endogenous *hlh-8* transcriptional reporter using the strategy described in Luo et al. [37]. Using CRISPR, we inserted the intergenic trans-splicing acceptor region from CEOPX036 followed by *nls::gfp::nls* at the end of the *hlh-8* coding region. The resulting *hlh-8 (jj422[hlh-8::sl2::nls::gfp::nls])* (denoted *jj422[hlh-8p::gfp]*) is a bicistronic allele where *hlh-8* and *gfp* are co-transcribed under the *hlh-8* promoter (Fig 6A), and expression of nuclear localized GFP is indicative of endogenous *hlh-8* expression. Animals expressing this endogenous *hlh-8* transcriptional reporter do not exhibit any overt phenotypes exhibited by *hlh-8(0)* mutants [12], suggesting that *hlh-8* function is not compromised in *jj422[hlh-8p::gfp]* animals. The GFP signal of this endogenous *hlh-8* transcriptional reporter is rather faint, yet it has the same expression pattern as the transgenic *jjIs3900[hlh-8p::nls::mCherry]* and *ayIs6[hlh-8p::gfp]* reporters in the M lineage (Fig 6B–6D'').

We then examined the expression of the endogenous *hlh-8* transcriptional reporter in *sem-2 (jj321[P158S])* mutants. The expression of *jj422[hlh-8p::gfp]* in the early M lineage is too faint to make accurate comparisons between WT and *sem-2(jj321[P158S])*. At the SM stage, the expression level of *jj422[hlh-8p::gfp]* was significantly reduced in at least one of the two SMs in *sem-2 (jj321[P158S])* mutants (81.0%, N = 42) compared with wild-type animals that express *sem-2* in both SMs (94.3%, N = 35) (Fig 6E–6H). Similarly, in 50% of *sem-2(jj321[P158S])* animals, *jjIs3900 [hlh-8p::nls::mCherry]* is undetectable in one of two SMs (N = 26), whereas wild-type animals always have detectable expression in the SMs (100%, N = 24) (S2F–S2I'' Fig). We quantified the expression of *jj422[hlh-8p::gfp]* in the *sem-2(jj321[P158S])* SMs that expressed it, and found that *sem-2(jj321[P158S])* animals exhibit reduced expression of *jj422[hlh-8p::gfp]* compared to wild-type animals (Fig 6I). Thus, based on data using both the transgenic and endogenous *hlh-8* transcriptional reporters, SEM-2 is required to regulate *hlh-8* expression in the M lineage.

## *sem-2(P158S)* mutants exhibit abnormal expression of several HLH-8 direct target genes and have defects in SM proliferation and egg-laying muscle differentiation

HLH-8 is known to directly, but differentially, regulate the expression of several reporter genes in the M lineage, including *egl-15p::gfp* (expressed in vm1s), *arg-1p::gfp* (expressed in all vms),

**Fig 6. SEM-2 is required for endogenous *hlh-8* expression in the M lineage.** A) Schematic representation of an endogenous *hlh-8* transcriptional reporter generated by introducing a sl2 spliced leader sequence from CEOPX036 and the sequence of *nls::gfp::nls* at the end of the HLH-8 coding region. This reporter *hlh-8(jj422[hlh-8::sl2::nls::gfp::nls])* is denoted *jj422[hlh-8p::gfp]*. Gray boxes represent *hlh-8* exons with introns separating them. The orange color indicates the sl2 sequence, the blue color indicates the nuclear localization signal (*nls*), and the green color indicates *gfp*. B–D") Fluorescence images showing the expression of *jj422[hlh-8p::gfp]* (B'–D'), the transgenic *jjIs3900[hlh-8p::nls::mCherry]* (B–D), and corresponding merged images (B"–D") in the M lineage at the 1-M (B–B"), 2-SM (C–C"), and 8-SM (D–D") stages. *jj422[hlh-8p::gfp]* has the same expression pattern as the transgenic *jjIs3900[hlh-8p::nls::mCherry]*, but the *jj422[hlh-8p::gfp]* signal in the M mesoblast cell is faint. Arrow in B' points to the M mesoblast cell. E–H) Fluorescence images (E'–G') of wild-type (E–E") and *sem-2(jj321[P158S])* (F–G") L3 animals showing expression of *jj422[hlh-8p::gfp]* in SMs labelled by the transgenic *hlh-8p::nls::mCherry* reporter (E–G). Merged images are shown in E"–G". Scale bars represent 10 μm. H) Quantification of the number of SMs in wild-type and *sem-2(jj321[P158S])* animals that express *jj422[hlh-8p::gfp]*. I) Quantification of GFP fluorescence intensity in *jj422[hlh-8p::gfp]*-expressing SM cells in WT and *sem-2(jj321[P158S])*. Each dot represents an animal scored. Data are normalized to WT. Statistical significance was calculated by performing unpaired two-tailed Student's *t*-tests. * $P < 0.05$.

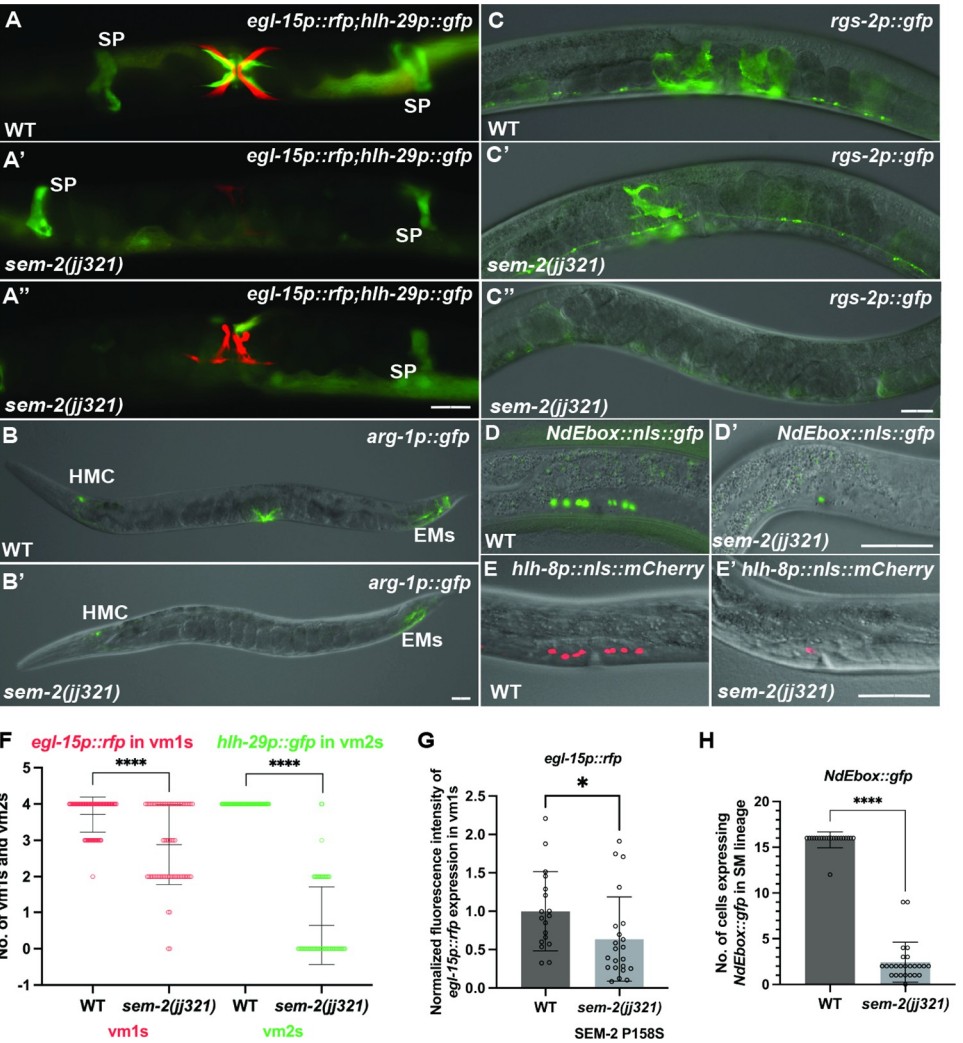

**Fig 7. *sem-2(P158S)* mutants have reduced numbers and deformed egg-laying muscles.** A–E') Images of wild-type (A–E) and *sem-2(jj321[P158S])* mutant (A'–E') gravid adult animals (A–C") or L4 animals (D–E') showing the expression of *egl-15p::rfp* and *hlh-29p::gfp* (A'–A"), *arg-1p::gfp* (B–B'), *rgs-2p::gfp* (C–C"), *NdEbox::gfp* (D–D') and *hlh-8p::nls::mCherry* (E–E'). *sem-2(jj321[P158S])* mutants either have no (A') or a reduced number of egg-laying muscles expressing *hlh-29p::gfp* (A"), with no change in *hlh-29p::gfp* expression in the spermatheca (SP). *sem-2(jj321[P158S])* mutants also have either reduced expression (A') or a reduced number of egg-laying muscles expressing *egl-15p::gfp* (A"). *arg-1p::gfp* expression is completely lost in the vms, but remains in the head mesodermal cell (HMC) and the enteric muscles (EMs), of *sem-2(jj321[P158S])* mutants (compare B and B'). *sem-2(jj321[P158S])* mutants also have either no expression (C") or a reduced number of cells (C') expressing the um marker *rgs-2p::gfp*, with no change in its expression in the nerve cord. At the L4 larval stage, *sem-2(jj321[P158S])* mutants have reduced number of cells expressing *NdEbox::gfp* and *hlh-8p::nls::mCherry* (D'–E') compared to WT (D–E). F–H) Graphs showing the number of vm1s and vm2s as indicated by the expression of *egl-15p::rfp* (vm1s) and *hlh-29p::gfp* (vm2s) (F), the expression level of *egl-15p::rfp* (G), and the number of *Ndebox::gfp*-expressing cells (H) in wild-type and *sem-2(jj321[P158S])* animals. Each dot represents an animal scored. Data are normalized to WT in panel G. Statistical significance was calculated by performing unpaired two-tailed Student's *t*-tests. **** *P*<0.0001. * *P*<0.05, Scale bars represent 40 μm.

and *NdEbox::gfp* (expressed in all vms and ums) [9–11,38,39]. Because *hlh-8* expression is reduced in *sem-2(P158S)* mutants, we asked whether these *sem-2* mutants exhibited altered expression of HLH-8 target genes. As shown in Fig 7A–7B', while 100% WT animals had *arg-1p::gfp* expression in vms (N = 30) or *egl-15p::gfp* in vm1s (N = 100), only 1.6% of *sem-2(jj321 [P158S])* mutants examined (N = 63) expressed *arg-1p::gfp* in the vms. Further, while 95% *sem-*

2(jj321[P158S]) mutants expressed the vm1 marker *egl-15p::rfp* (N = 138) (Fig 7A'–7A"), there was a significant reduction of the number of *egl-15p::rfp*-positive cells in s*em-2(jj321[P158S])* mutants (Fig 7A" and 7F), and these cells appeared deformed (Fig 7A–7A"). Moreover, the fluorescence intensity of *egl-15p::rfp* in s*em-2(jj321[P158S])* mutants was slightly reduced compared to WT animals (Fig 7G). In addition, fewer cells in *sem-2(jj321[P158S])* mutants expressed *NdEbox::gfp*, although the expression level of *Ndebox::gfp* appeared unchanged in the cells that expressed it (100%, N = 25) (Figs 7D–7D' and 7H). Thus, *sem-2(P158S)* mutants exhibit abnormal expression of target genes directly regulated by HLH-8.

HLH-8 is known to regulate the proper proliferation of the SMs, and the proper differentiation and function of the vulval muscles [11,12]. Consistent with a role of SEM-2 in regulating the expression of *hlh-8*, *sem-2(P158S)* mutants also exhibited defects in SM proliferation and vulval and uterine muscle differentiation. While *sem-2(jj321[P158S])* mutants have two SMs, as indicated by the expression of the endogenously tagged GFP::2xFLAG::SEM-2 in *sem-2 (jj382 jj417[GFP::2xFLAG::SEM-2(P158S)]* animals (92%, N = 25) (S2H–S2I" Fig), the SMs do not divide at the wild-type rate (Fig 8A–8G). At the young to mid- L4 larval stage when wild-type animals have 16 SM descendants, *sem-2(jj417[P158S])* mutants have fewer than 16 cells resembling SMs in the developing vulva region (Figs 8A–8G and 7E–E'). Consistent with an SM proliferation defect, only 28% of *sem-2(jj321[P158S])* mutants (N = 139) expressed the vm2 marker *hlh-29p::gfp* [40], and 50% of *sem-2(jj321[P158S])* mutants (N = 20) expressed the um marker *rgs-2p::gfp* [41]. We further monitored the M lineage division patterns in WT and *sem-2(P158S)* mutants and found that the mutants exhibit variable patterns of reduced SM proliferation. Fig 8G summarizes the different types of SM division patterns observed in *sem-2 (jj321[P158S])* mutants. Thus, SEM-2 is required for the proper proliferation of the SMs and the differentiation of the various egg-laying muscles.

## A putative SoxC-binding site is critical for *hlh-8* promoter activity in the M lineage

To determine if SEM-2 directly regulates *hlh-8* expression and to also identify regions in the *hlh-8* promoter that are important for expression in the M lineage, we performed transgenic reporter assays by generating a series of deletion constructs in the *hlh-8* promoter and testing the ability of each deleted *hlh-8* promoter to drive GFP expression in the M lineage. As shown in Fig 9A–9H", we uncovered two 20bp regions required for *hlh-8* promoter activity in the M lineage: one between -280bp and -260bp, which we named E1, and another between -220bp and -200bp, which we named E2 (Figs 9A–9H" and S5A). E1 is highly conserved among multiple nematode species (S5A Fig), and it contains a putative Sox transcription factor binding site, which we named Site1, based on the transcription factor binding site identifier PROMO (https://alggen.lsi.upc.edu). The consensus sequence for SoxC binding is CA/TTTGTT (S5B Fig) [13, 42, 43]. Site1 contains the exact sequence for SoxC binding AACAAAGaagaag and is located at -272bp to -259bp upstream of the *hlh-8* start codon (S5 Fig). E2 has a sequence that matches the consensus sequence for SoxC binding by 6 out of 7 nucleotides (CTTTCTTttc), which we named Site2. Site2 is located at -221bp to -211bp upstream of the *hlh-8* start codon (S5 Fig).

We then tested the importance of the putative SoxC-binding sites in Site1 and Site2 in vivo, by mutating Site1 alone (because it contains an exact match to the SoxC binding consensus) or Site1 and Site2 together in the endogenous *hlh-8* transcriptional reporter background, *jj422 [hlh-8p::gfp]*. We generated three alleles: *jj483 jj422 [hlh-8p(Site1m+Site2m)::gfp]*, which has a 13bp mutation with the putative SoxC-binding site in Site1 mutated and a 10bp mutation with the putative SoxC-binding site in Site2 mutated, as well as *jj445 jj422 [hlh-8p(Site1m)::gfp]* and

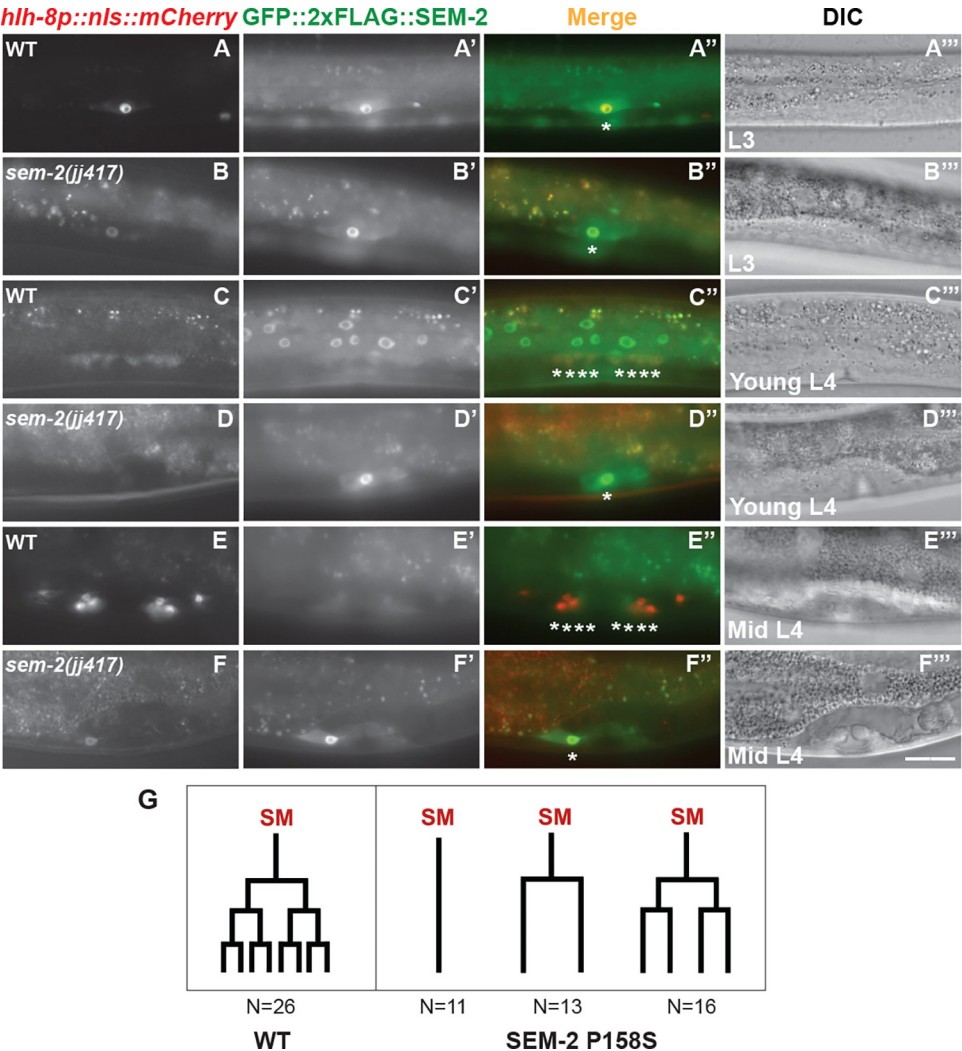

**Fig 8. *sem-2(P158S)* mutants have SM proliferation defects.** Fluorescence and DIC images showing SMs and SM descendants labelled by *hlh-8p::nls::mCherry* (A–F) and GFP::2xFLAG::SEM-2 (A'–F') in wild-type *(sem-2(jj382 [GFP::2xFLAG::SEM-2]))* (A–A"', C–C"', E–E"') and *sem-2(P158S)* *(sem-2(jj382 jj417[GFP::2xFLAG::SEM-2(P158S)]))* mutant (B–B"', D–D"', F–F"') animals at the L3 (A–B"'), young L4 (C–D"'), and mid L4 (E–F"') stages. Corresponding merged and DIC images are shown in A"–F" and A"'–F"', respectively. Asterisks label SMs and SM descendants. Scale bar represents 20 µm. G) Schematic representations of the SM lineage division patterns in WT and *sem-2(P158S)* mutants. Total number of animals scored was 26 for WT and 40 for *sem-2(P158S)*.

*jj446 jj442 [hlh-8p(Site1m)::gfp]*, both of which have a 13bp mutation with the putative SoxC-binding site in Site1 mutated (Fig 10A, S1 and S3 Tables). All three alleles resulted in reduced or undetectable levels of GFP in the SMs: *jj483 jj422 [hlh-8p(Site1m+Site2m)::gfp]* have undetectable levels of GFP in 97.4% of SMs scored (N = 39), *jj445 jj422 [hlh-8p(Site1m)::gfp]* and *jj446 jj442 [hlh-8p(Site1m)::gfp]* have undetectable levels of GFP in 96.1% of SMs scored (N = 52). In contrast, wild-type animals have detectable levels of GFP in 99% of all the SMs scored (N = 96). The few animals carrying Site1 and Site2 mutations (*jj483*) or only Site 1 mutations (*jj445/jj446*) that expressed *jj422[hlh-8p::gfp]* in the SMs had significantly reduced expression compared to wild-type animals. These results indicate that the putative SoxC-binding site in Site1 is important for *hlh-8* expression in the M lineage. Intriguingly, *jj422[hlh-8p:: gfp]* expression at the 16-SM stage in the three mutants appeared comparable to that in wild-

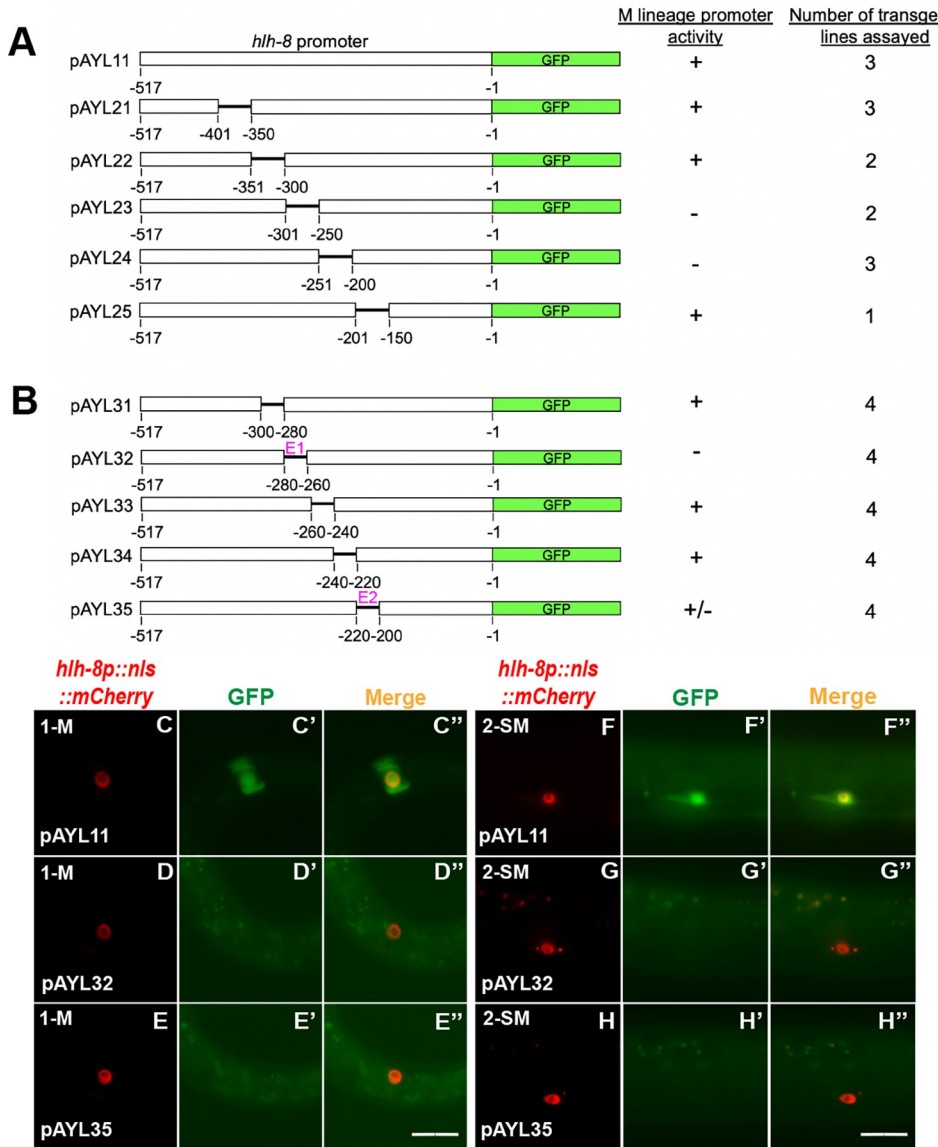

**Fig 9. Two 20bp regions in the *hlh-8* promoter are necessary for *hlh-8* promoter activity in the M lineage.** A) Schematics of the wild-type construct and 50bp deletions in a *hlh-8* transgenic transcriptional reporter that has 517bp of the *hlh-8* promoter located immediately upstream of its start codon driving the expression of *gfp*. B) Schematics of *hlh-8* transcriptional reporter constructs with 20bp deletions in a 100bp region located at -300bp to -200bp upstream of the start codon of *hlh-8*. E1 and E2 contain putative SEM-2/SoxC-binding sites and are indicated by the color magenta. C–E") Representative fluorescence images of reporter expression (C'–H') in the M mesoblast cell of L1 animals (C–E") and in SMs of L3 animals (F–H") labelled by the *hlh-8p::nls::mCherry* reporter (C–H). C"–H" are merged images. Scale bars represent 20 μm.

type animals (Fig 10E–10G"), suggesting that mutating the putative SoxC-binding sites significantly reduces, but does not completely abolish, *hlh-8* expression in the M lineage. Consistent with this notion, none of these three mutants exhibited an Egl phenotype, a *hlh-8* null-like phenotype. We reasoned that there might be additional SoxC-binding site(s) in the endogenous *hlh-8* genomic region that contributes to the activation of *hlh-8* expression in the absence of the two putative SoxC-binding sites in Site1 and Site2. Alternatively, *hlh-8* expression in the later stages of M lineage development does not depend on SEM-2.

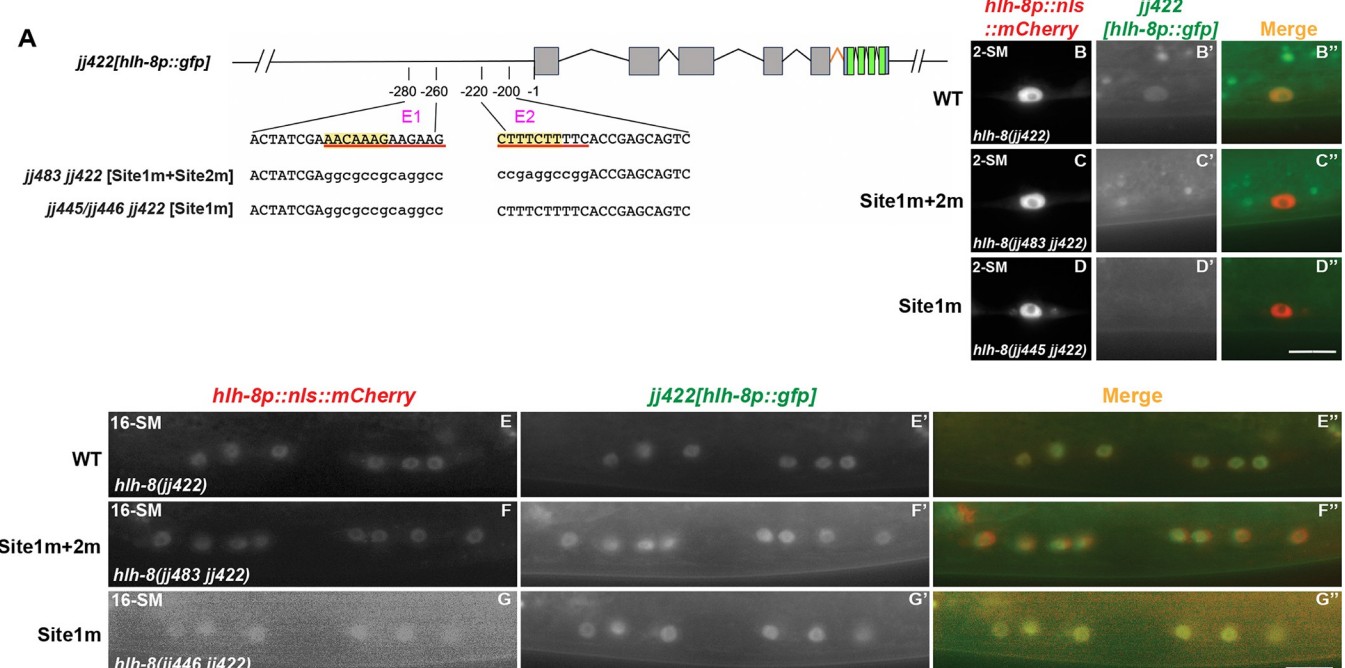

**Fig 10. A putative SEM-2/SoxC-binding site is important for robust endogenous *hlh-8* expression in the M lineage.** A) A schematic showing mutations made in two putative SEM-2/SoxC-binding sites in the *hlh-8* promoter in worms carrying an endogenous *hlh-8* transcriptional reporter, *jj422[hlh-8p::gfp]*. The sequences of E1 (20bp region identified in Fig 9) and E2 (20bp region identified in Fig 9) are listed. The putative SEM-2/SoxC-binding sites are highlighted in yellow. Site1 (13bp) and Site2 (10bp) are underlined in red. Mutations in Site1 and/or Site2 in (*hlh-8(jj483 jj422)*) and (*hlh-8(jj445/446 jj422)*) are indicated by lowercase letters. B–D") Fluorescence images (B'–D') of a wild-type *(hlh-8(jj422))* (B–B"), a Site1 and Site2 mutant *(hlh-8(jj483 jj422))* (C–C"), and a Site1 mutant *(hlh-8(jj445 jj422))* (D–D") L3 animal showing expression of *jj422[hlh-8p::gfp]* in SMs labelled by the *hlh-8p::nls::mCherry* reporter (B–D). Merged images are shown in B"–D". *jj483 jj422 [hlh-8p(Site1m+Site2m)::gfp]* have undetectable levels of GFP in 97.4% (N = 39) of SMs scored, *jj445 jj422 [hlh-8p (Site1m)::gfp] and jj446 jj442 [hlh-8p(Site1m)::gfp]* have undetectable levels of GFP in 96.1% of SMs scored (N = 52), while wild-type animals have detectable levels of GFP in 99% (N = 96) of SMs scored. E–G") Fluorescence images (E'–G') of a wild-type *(hlh-8(jj422))* (E–E"), a Site1 and Site2 mutant *(hlh-8(jj483 jj422))* (F–F"), and a Site1 mutant *(hlh-8(jj446 jj422))* (G–G") animals showing expression of *jj422[hlh-8p::gfp]* in SM descendants labelled by the *hlh-8p::nls:: mCherry* reporter (E–G). Merged images are shown in E"–G". Scale bars represent 10 μm.

## Discussion

By taking advantage of a partial loss-of-function allele of *sem-2*, we have identified additional functions of the single *C. elegans* SoxC protein, SEM-2, and uncovered previously unappreciated regulatory relationships between SEM-2 and LET-381/FoxF/C, as well as between SEM-2 and HLH-8/Twist. Our work adds new subcircuits to the gene regulatory network underlying *C. elegans* postembryonic development [2].

### SEM-2/SoxC functions antagonistically with LET-381/FoxF/C in CC fate specification

The zinc finger transcription factor SMA-9/Schnurri is known to regulate the expression of the forkhead transcription factor LET-381/FoxF/C, and both proteins are required for specifying the M-CC fate in the dorsal M lineage. Loss of either transcription factor results in a fate transformation of M-CCs to SMs due to de-repression of *sem-2* expression [8,24]. In this study, we have found that the *sem-2(P158S)* mutation, which renders the SEM-2 protein partially functional, can suppress the loss of M-CC phenotype of *sma-9(0)* mutants (Susm), and that this Susm phenotype is dependent on the presence of LET-381 (Tables 1 and 3, Fig 4). These findings support a role of SEM-2 in the dorsal side of the M lineage and a mutually

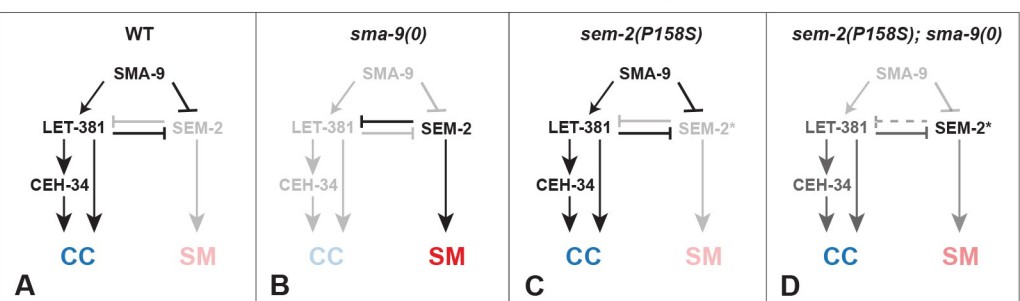

**Fig 11. A model on the regulatory network involved in CC specification in the dorsal M lineage.** In WT animals (A), LET-381 is expressed in the dorsal M lineage cells fated to become CCs due to the presence of SMA-9 in these cells, either by repressing the expression of *sem-2*, thus preventing SEM-2 from repressing the expression of LET-381, or by activating *let-381* expression independently of SEM-2. Once expressed, LET-381 further represses the expression of *sem-2*, while at the same time, directly activates the expression of CEH-34 and functions in a feedforward manner with CEH-34 to activate genes required for the differentiation and function of CCs. In *sma-9(0)* mutants (B), the de-repression of SEM-2 expression leads to the conversion of CCs to SMs as the fully functional SEM-2 represses *let-381* expression. The partially functional SEM-2 protein expressed in *sem-2(P158S)* mutants (SEM-2*) on its own (C) is not expected to affect the CC fate. However, in *sma-9(0)* mutants, the de-repressed expression of a partially functional SEM-2(P158S) protein (SEM-2*) is not sufficient to completely repress LET-381 expression, thus leading to a fraction of *sem-2(P158S); sma-9(0)* double mutants producing CCs in the dorsal M lineage (D). Dotted line in panel D indicates that the SEM-2(P158S) protein (SEM-2*) is not able to fully repress *let-381* expression.

repressive relationship between SEM-2 and LET-381 (Fig 11A). We therefore propose a model shown in Fig 11A. Based on this model, in wild-type animals, SMA-9 functions, either acting through or independently of LET-381, to repress *sem-2* expression in the presumptive CC mothers (M.dlp and M.drp) and CCs (M.dlpa and M.drpa) in the dorsal M lineage (Fig 11A). The expression of *let-381* in M.dlpa and M.drpa then leads to the activation of *ceh-34* expression, where LET-381 and CEH-34 function in a feedforward manner to regulate M-CC specification and function [8]. In *sma-9(0)* single mutants, ectopic expression of a fully functional *sem-2* in the dorsal M lineage leads to the repression of *let-381* expression in the presumptive CCs, causing the transformation of these cells to SMs (Fig 11B). *sem-2(P158S)* single mutants look like wild-type animals regarding CC specification, because of the actions of SMA-9 and LET-381 in preventing *sem-2* expression in the dorsal M lineage (Fig 11C). In *sem-2(P158S); sma-9(0)* double mutants, the partially functional SEM-2(P158S) protein being expressed in the dorsal side of the M lineage is not sufficient to fully repress *let-381* expression, leading to the expression of *let-381* and the formation of M-CCs in a fraction of these double mutant animals (Fig 11D). This model is consistent with our previous findings that forced expression of *sem-2* throughout the M lineage leads to the conversion of M derived-CCs and BWMS to SMs [6]. Moreover, it adds the possibility that SMA-9 activates *let-381* expression by way of repressing *sem-2* expression, forming a double negative gate. Future work will aim to determine whether SEM-2 and LET-381 directly regulate each other's expression in the M lineage.

The repression of *sem-2* expression in the M lineage by SMA-9 appears to be stage-specific, as *sem-2* expression in the early M lineage (1-M stage to 8-M stage) does not change in *sma-9(0)* mutants (S3 Fig). SMA-9 does not appear to be the only factor repressing *sem-2* expression in the non-SM cells at the 16- to 18-M stage. We have previously shown that HLH-1 and FOZI-1 repress *sem-2* expression in the M-derived BWMs, and that this genetic interaction is reciprocal, as SEM-2 is known to repress the expression of *hlh-1* and *fozi-1* in the SM mother cells and the SMs [6]. Similarly, expression of *sem-2* in the SM mother cells and the SMs requires LIN-12/Notch signaling and the zinc finger transcription factor SEM-4 [6,44], implicating additional levels of complexity in the regulatory network underlying proper fate

specification in the M lineage. It is clear that the specification of M-CCs and SMs involves intricate gene regulatory networks that include both positive and negative feedback, feedforward, and mutually antagonistic regulatory subcircuits. As previously suggested [45], such regulatory logic ensures temporal and spatial specificity of gene expression and robust cell fate specification.

## SEM-2/SoxC is necessary for the specification, proliferation, and differentiation of the SMs and SM descendants

Previous studies have shown that SEM-2 specifies the multipotent and proliferative SM fate [6]. Since the SMs are not made in *sem-2(n1343)* mutants that were used in the previous study, we could not determine if SEM-2 plays a role beyond SM specification in the SM lineage. In most *sem-2(P158S)* mutants, the level of functional SEM-2 in the M lineage is sufficient to specify the SM fate (92%, N = 25) (S2F–S2I" Fig). However, the SMs in *sem-2(P158S)* mutants exhibit reduced proliferation. These results provide direct evidence supporting a role of SEM-2 in regulating cell proliferation. It is possible that SEM-2 functions upstream of, and/or works cooperatively with, certain cell cycle regulators, either through positive regulation of G1/S-phase cyclins (Cyclin D/CDK4, Cyclin E/CDK2) or negative regulation of CDK inhibitors like CKIs (CKI-1/CKI-2) [46–51]. This role of SEM-2 in regulating cell proliferation appears to be evolutionarily conserved. Sox4, one of the SoxC proteins in humans, is often amplified and overexpressed in multiple cancers, and Sox4 is known to play crucial roles in cancer development and progression, and has been classified as a "cancer signature" gene [16,18].

In addition to regulating SM specification and proliferation, SEM-2 is essential for the proper differentiation of multiple non-striated muscle cells derived from the SM lineage. We have shown that while the vm1-specific reporter (*egl-15p*::*rfp*) is expressed in over 90% of *sem-2(P158S)* mutants, a significantly smaller fraction of *sem-2(P158S)* mutant animals express the vm2-specific reporter (*hlh-29p*::*gfp*, 28%), the um-specific marker (*rgs-2p*::*gfp*, 50%), or another vm marker (*arg-1p*::*gfp*, 1.6%) (Fig 7). These findings suggest that SEM-2 is important for the proper differentiation of the various non-striated muscles derived from the SM lineage. The more prevalent expression of the vm1-specific reporter (*egl-15p*::*rfp*) in *sem-2(P158S)* mutants is similar to our previous report showing that in *lin-39(0) mab-5(0)* mutants where the M lineage exhibits reduced cell proliferation, the few M lineage cells precociously differentiate to express the vm1 marker *egl-15p*::*gfp*, but not vm2 or um markers [52]. This shared, reduced proliferation phenotype by *lin-39(0) mab-5(0)* mutants and *sem-2(P158S)* mutants is consistent with MAB-5 and LIN-39 directly activating the expression of SEM-2 in the M lineage [6].

## The SoxC-Twist axis as a conserved regulatory cassette in metazoan development

Multiple lines of evidence support the role of SEM-2 in regulating the expression of *hlh-8/Twist* in the M lineage, possibly directly. First, both transgenic *hlh-8* transcriptional reporters (*hlh-8p*::*gfp* and *hlh-8p*::*nls*::*mCherry*) and an endogenous *hlh-8* transcriptional reporter all exhibited significantly reduced expression in *sem-2(P158S)* mutants (Figs 5 and 6). Moreover, the *sem-2(P158S)* mutant phenotype is similar to the *hlh-8* mutant phenotype. Both *sem-2(P158S)* mutants and *hlh-8* null, *nr2061*, mutants are Egl due to missing vulval muscles (this study, [12]). Additionally, animals with a semidominant allele of *hlh-8*, *n2170* (E29K), have SMs that often fail to divide [11], a phenotype similar to the SM proliferation defect in the *sem-2(P158S)* mutants. Finally, *sem-2(P158S)* mutants exhibit reduced, yet differential, expression of several HLH-8 target genes, such as *egl-15p*::*gfp* and *arg-1p*::*gfp*, a phenotype that has

been previously observed in various *hlh-8* mutants. For example, several *hlh-8* point mutants (R103M, R103A, L95F and F99L) express *egl-15p::gfp*, but do not express *arg-1p::gfp*, in the vulval muscles [39]. Similarly, worms containing *hlh-8(tm726)*, a 646-nucleotide deletion at the 3' end of intron one, express *egl-15p::gfp* (15% of animals) but do not express *arg-1p::gfp* [10], whereas animals that are heterozygous for *hlh-8(n2170)* express *arg-1p::gfp* but do not express *egl-15p::gfp* [11]. These results are consistent with altered or reduced endogenous *hlh-8* expression in *sem-2(P158S)* mutants.

It is likely that SEM-2 directly regulates the expression of *hlh-8* in the M lineage, through at least one of the putative SoxC-binding sites (Site1) in the *hlh-8* promoter (Figs 10A and S5). We have shown that E1, a 20bp region containing Site1, is essential for *hlh-8* promoter activity in transgenic reporter assays, and Site1 is important for *hlh-8* expression in the endogenous genomic environment (Figs 9 and 10). Mutating the putative SoxC-binding sites in Site1 and Site2 significantly reduced, but did not completely abolish, *hlh-8* expression in the endogenous locus, possibly due to the presence of other putative SoxC-binding sites in the *hlh-8* genomic region. Nevertheless, our results collectively are consistent with SEM-2 playing an important, likely a direct, role in regulating the expression of *hlh-8* in the M lineage.

In humans Twist1, Twist2, FGFRs and JAG-1/Notch2 are known to play important roles in craniofacial development [53, 54]. The *C. elegans* homolog of Twist1/2 is HLH-8, the homolog of the FGFRs is EGL-15, and the homolog of JAG-1 is ARG-1 [11,39,55]. *egl-15* and *arg-1* are direct targets of HLH-8, and they have each been shown to be expressed in and/or to work in patterning a subset of mesodermal tissues: the egg-laying muscles and the enteric muscles [11,12]. Mutations in *hlh-8* and *egl-15* lead to Egl and/or Con phenotypes, which have been labelled as phenologs of craniofacial defects in humans [39]. The *sem-2(P158S)* mutants are 100% Egl (Fig 1G–1H, Table 1). Intriguingly, mutations in SoxC proteins, Sox4 and Sox11, are associated with a developmental disorder called Coffin-Siris syndrome (CSS), and one key characteristic of CSS patients is craniofacial defects [19–21]. Similarly, SoxC proteins are known to function upstream of Twist1, in some cases directly, in disease initiation and progression in mammals, particularly in the regulation of epithelial-mesenchymal transition (EMT) [16,56–58]. Thus, the SoxC-Twist axis, including the downstream targets of Twist, such as FGFRs and JAG-1, represents an evolutionarily conserved regulatory cassette important in metazoan development.

## Supporting information

**S1 Table. *C. elegans* strains used in this study.**
(DOCX)

**S2 Table. Plasmids generated in this study.**
(DOCX)

**S3 Table. Oligonucleotides used in this study.**
(DOCX)

**S1 Fig. Quantification of GFP::2xFLAG::SEM-2 expression during M lineage development.** A–C) Quantification of GFP::2xFLAG::SEM-2 fluorescence intensity in the nuclei (A, D, G), the cytoplasm (B, E, H), and the ratio of nuclear signal to cytoplasmic signal (C, F, I) in the early M lineage (A, B, C), the SM lineage (D, E, F), and throughout M lineage development (G, H, I). All images were taken at the same exposure and same magnification. Each dot represents a cell scored. Data for GFP::2xFLAG::SEM-2 expression at the 16-M stage in M-derived BWMs and SM mother cells are denoted 16-M(B) and 16-M(S), respectively. For graphs D-I, only data for the SMs at the 18-M stage are shown. For panels C, F, and I, the ratios were

calculated by dividing the nuclear GFP intensity by the cytoplasmic GFP intensity. For panels A–C, data were normalized to the 1-M stage. For panels D–I, data were normalized to the 18-M stage. Statistical analysis was done using one-way ANOVA with Dunnett's test. **** $P < 0.0001$, *** $P < 0.001$, ** $P < 0.01$, * $P < 0.05$, ns, not significant. All the corresponding data shown in panels A-C and D-F were combined and shown in panels G-I. To prevent the graph from being too crowded, only $P$ values not shown in panels A-F are shown in G-I. There is a gradual decrease in the level of nuclear GFP::2xFLAG::SEM-2 in the early M lineage after the 2-M stage (A). At the 16-M stage, there is an upregulation of nuclear GFP::2xFLAG::SEM-2 in the SM mothers (A). At the 18-M stage, this increase in nuclear GFP::2xFLAG::SEM-2 persists in the SMs but becomes undetectable in the BWMs (including in the SM sister cells, which transiently express GFP::2xFLAG::SEM-2) upon terminal differentiation. There is then a gradual decrease in the level of nuclear GFP::2xFLAG::SEM-2 signal in the SM descendants (D). The level of cytoplasmic GFP::2xFLAG::SEM-2 signal appears relatively stable within the early M lineage (B), and within the SM lineage (E). The ratio of nuclear to cytoplasmic localization of GFP::2xFLAG::SEM-2 in the M lineage follows the pattern of change of nuclear GFP::2xFLAG::SEM-2 (C, F, I). There is an increase in the level of nuclear and cytoplasmic GFP::2xFLAG::SEM-2, as well as their ratio, in the SMs at the 18-M stage compared with the M mesoblast cell at the 1-M stage (G, H, I).
(TIF)

**S2 Fig. Quantification of GFP::2xFLAG::SEM-2 expression in the M lineage of wild-type and SEM-2 P158S mutant animals.** A–B") Fluorescence images showing GFP::2xFLAG:: SEM-2 (A'–B') in the M mesoblast cell labelled by the *hlh-8p::nls::mCherry* reporter (A–B) at the 1-M stage in WT (A–A") and *sem-2(jj417[SEM-2 P158S])* (B–B") hermaphrodites. (A"–B") are the corresponding merged images. The GFP::2xFLAG::SEM-2 images were taken at the same exposure and same magnification. C–E) Quantification of GFP::2xFLAG::SEM-2 in the nuclei (C), the cytoplasm (D), and the ratio of nuclear to cytoplasmic signal (E) in the M mesoblast cell of wild-type and *sem-2(jj417[SEM-2 P158S])* animals. F–I") Fluorescence images of a wild-type (*sem-2(jj382)*) animal (F–G") and a *sem-2(jj417[P158S])* mutant animal (H–I"), showing GFP::2xFLAG::SEM-2 (F'–I'), *hlh-8p::nls::mCherry* (F–I), and the corresponding merged images (F"–I") in the two SMs. All GFP::2xFLAG::SEM-2 images were taken at the same exposure and same magnification, while *hlh-8p::nls::mCherry* images in H and I were taken using a longer exposure than those in F and J. J–L) Quantification of GFP::2xFLAG:: SEM-2 in the nuclei (J), the cytoplasm (K), and the ratio of nuclear to cytoplasmic signal (L) in the SMs of wild-type and *sem-2(jj417[SEM-2 P158S])* animals. For panels E and L, the ratios were calculated by dividing the nuclear GFP intensity by the cytoplasmic GFP intensity. Each dot represents a cell scored. Data are normalized to WT. Statistical significance was calculated by performing unpaired two-tailed Student's *t*-tests. *** $P < 0.001$, * $P < 0.05$, ns, not significant. Scale bars represent 10 μm. Arrows point to the M mesoblast cell, while arrowheads point to the SM cell.
(TIF)

**S3 Fig. *sem-2* expression in the early M lineage does not change in *sma-9(0)* mutants.** A–D") Fluorescence images showing GFP::2xFLAG::SEM-2 (A'–D') in M lineage cells labelled by the *hlh-8p::nls::mCherry* reporter (A–D) at the 2-M stage (A–B") and 8-M stage (C–D") of M lineage development in WT (A–A", C–C") and *sma-9(0)* (B–B", D–D") hermaphrodites. (A"–D") are the corresponding merged images. Only the left side of an animal is shown in this figure, while the other side is out of the focal plane. Scale bar represents 20 μm. E–G) Quantification of GFP::2xFLAG::SEM-2 in the nuclei (E), the cytoplasm (F), and the ratio of nuclear to cytoplasmic signal (G) in the early M lineage of WT and *sma-9(0)* mutants. For panel G, the

ratios were calculated by dividing the nuclear GFP intensity by the cytoplasmic GFP intensity. Each dot represents a cell scored. Data are normalized to WT at the 1–2 M stage. Statistical significance was calculated by performing unpaired two-tailed Student's *t*-tests. ns, not significant. (TIF)

**S4 Fig. The regulation of *hlh-8* by SEM-2 is not transgene- or *sem-2* allele-specific.** A–B") Fluorescence images (A–B) of wild-type (A–A") and *sem-2(jj321[P158S])* (B–B") embryos showing the expression of *hlh-8p::nls::mCherry* in the M mesoblast cell (arrows). A'–B' and A"–B" are corresponding DIC and merged images, respectively. Transgenic animals expressing *hlh-8::nls::mCherry* were generated with a co-injection marker *myo-2p::mCherry* represented by the red pharyngeal signal. Scale bar represents 15 μm. C) Quantification of *hlh-8p::nls::mCherry* expression in the M mesoblast cell of WT and *sem-2(jj321[P158S])* embryos. Each dot represents an embryo scored. Data are normalized to WT. Statistical significance was calculated by performing unpaired two-tailed Student's *t*-tests. * *P*<0.05. D–E) Fluorescence images of wild-type (D–D") and *sem-2(jj152[P158S])* (E–E") mutant L3 animals showing expression of the *hlh-8p::gfp* transgene in SMs. D'–E' and D"–E" are corresponding DIC and merged images, respectively. Exposure for panel E is 20x times higher than panel D (1x). Scale bar represents 10 μm. F–G) Fluorescence images of wild-type (F) and *sem-2(jj152[P158S])* (G) mutant L1 animals at the 4-M stage showing expression of the *hlh-8p::gfp* transgene at the same exposure (1x). Scale bar represents 20 μm. H–I") Fluorescence images (H–I) of a heterozygous *sem-2* null (*sem-2(ok2422)/hT2[qIs48]*) (H–H") and a *sem-2* null (*sem-2(ok2422))* (I–I") embryo showing expression of *hlh-8p::nls::mCherry*. *myo-2p::gfp* images from the *hT2[qIs48]* balancer chromosome are shown in H'–I' and merged images are shown in H"–I". Scale bar represents 15 μm. J) Quantification of *hlh-8p::nls::mCherry* expression in the embryonic M mesoblast cell of *sem-2(ok2422)/hT2[qIs48]* and *sem-2(ok2422)* animals. Each dot represents an embryo scored. Data are normalized to *sem-2(ok2422)/hT2[qIs48]*. Statistical significance was calculated by performing unpaired two-tailed Student's *t*-tests. ns, not significant. K–L"') Fluorescence images (K–L) of a heterozygous *sem-2* null (*sem-2(ok2422)/hT2[qIs48]*) (K–K"') and a *sem-2* null (*sem-2(ok2422))* (L–L"') L1 animals showing expression of *hlh-8p::nls::mCherry*. *myo-2p::gfp* images from the *hT2[qIs48]* balancer chromosome are shown in K'–L', merged images of *hlh-8p::nls::mCherry* and *myo-2p::gfp* are shown in K"–L", and DIC images are shown in K"'–L"'. Scale bar represents 20 μm. Arrows point to M lineage cells. (TIF)

**S5 Fig. The putative SEM-2/SoxC-binding site in E1 is conserved.** A) Screenshot of the University of California, Santa Cruz (UCSC) genome browser showing the sequence conservation of E1 and E2 in the *hlh-8* promoter among 26 nematode species and a broader 135 species (112 nematodes, 22 flatworms, and *Ciona intestinalis)*. The magenta color highlights E1 and E2. Red boxes show Site1 and Site2. The yellow color highlights the putative SEM-2/SoxC-binding sites in E1/Site1 and E2/Site2. B) The position weight matrix of the Sox4/SoxC primary motif as determined by the protein-binding microarray (PBM) method [33]. (TIF)

**S1 Data. Excel file containing all the raw data for Figs 2, 5, 6, 7, S1, S2, S3 and S4.** (XLSX)

## Acknowledgments

We thank Yoko Takashima for generating the *sem-2(jj321); arg-1p::gfp* strain, Josh Arribere, Dan Dickinson, Andy Fire, Bob Goldstein and Oliver Hobert for plasmids, Sijung Yun for

analyzing whole genome sequencing data, Gunther Hollopeter for sharing CRISPR protocol, Yuxin Mao for advice on structural modeling, Peter Schweitzer and the Cornell Genomics Facility for help with whole genome sequencing assays, and members of the Liu lab for helpful discussions and critical comments on the manuscript.

## Author Contributions

**Conceptualization:** Marissa Baccas, Jun Liu.

**Data curation:** Marissa Baccas, Jun Liu.

**Formal analysis:** Marissa Baccas, Jun Liu.

**Funding acquisition:** Jun Liu.

**Investigation:** Marissa Baccas, Vanathi Ganesan, Amy Leung, Lucas R. Pineiro, Alexandra N. McKillop, Jun Liu.

**Project administration:** Jun Liu.

**Supervision:** Jun Liu.

**Validation:** Marissa Baccas.

**Visualization:** Marissa Baccas.

**Writing – original draft:** Marissa Baccas, Jun Liu.

**Writing – review & editing:** Marissa Baccas, Jun Liu.

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
