## [Decision Letter · Decision Letter 0]

22 Aug 2024

Dear Dr Liu,

Thank you very much for submitting your Research Article entitled 'SEM-2/SoxC regulates multiple aspects of C. elegans postembryonic mesoderm development' to PLOS Genetics.

The manuscript was fully evaluated at the editorial level and by independent peer reviewers. The reviewers appreciated the attention to an important problem, but raised some substantial concerns about the current manuscript. Based on the reviews, we will not be able to accept this version of the manuscript, but we would be willing to review a substantially-revised version. We cannot, of course, promise publication at that time.

Should you decide to revise the manuscript for further consideration here, your revisions should address the specific points made by each reviewer. In particular, while the work is well quantified in parts, other parts of the manuscript makes interpretations about relative reporter fluorescence intensity without quantification of intensity, as noted by reviewer 2. The authors should either further quantify the data and provide intensity measurements with statistical analysis in the relevant figures, or change the language of the claims. This is not necessarily needed when signal is stated to be undetectable, but is important when signal is detectable and claimed to be dimmer or brighter than controls. Reviewers 1 and 3 note areas that require better explanation and revision, such as the section describing promoter deletion analysis. We will also require a detailed list of your responses to the review comments and a description of the changes you have made in the manuscript.

If you decide to revise the manuscript for further consideration at PLOS Genetics, please aim to resubmit within the next 60 days, unless it will take extra time to address the concerns of the reviewers, in which case we would appreciate an expected resubmission date by email to plosgenetics@plos.org.

To resubmit, log into your Editorial Manager account and select the option 'Revise Submission' in the 'Submissions Needing Revision' folder.

Please do not hesitate to contact us if you have any concerns or questions.

Yours sincerely,

Barth D Grant

Guest Editor

PLOS Genetics

Giovanni Bosco

Section Editor

PLOS Genetics

Reviewer's Responses to Questions

**Comments to the Authors:**

Reviewer #1: This paper describes new roles for the C. elegans SoxC protein SEM-2 during development of a postembryonic mesodermal lineage. The basis of the work is a new allele of sem-2, which is a missense mutation in the DNA binding domain that results in a partial loss of function (by genetic criteria). The authors use this mutation to further explore the function of sem-2 in M lineage specification, adding to their already significant understanding of the gene regulatory networks governing M development. They find that SEM-2 is required for the proliferation and diversification of the SM lineage, acts antagonistically with LET-381/FoxF, and (likely directly) regulates hlh-8/Twist.

This work advances our understanding of the gene regulatory mechanisms governing development of the M mesoblast, which is a postembryonic multipotent progenitor cell. This gene regulatory network may be conserved in metazoans, as homologous genes have been implicated in craniofacial development in humans. The paper is well written. The results are strong and clearly support the conclusions.

Specific points:

1. Line 64. The M mesoblast is described as pluripotent. Pluripotency is the ability to develop into all three germ layers. M is more accurately described as multipotent.

2. Lines 356-359. It wasn’t clear to me why both jj321 and jj152 were examined, as both mutations result in the same amino acid change and all evidence suggests they have the same phenotype.

3. Lines 380-383. This sentence should be revised: “We reasoned that the more robust expression of sem-2 in sem-2(ok2422) null embryos born from [heterozygous] parents is likely due to the maternal contribution of SEM-2 by [heterozygous] parents”. I believe the authors offer an explanation as to why the phenotype of ok2422 is less severe than that of jj152. This needs to be made clearer.

4. Figure 5G-M. It appears that there is more background for the hlh-8::mCherry images in the sem-2 mutant background (H, L’, M’). This may be due to a longer exposure that is required because hlh-8 expression is reduced in the mutants. Exposure differences should be indicated in the figure legend.

5. Lines 485-487. An alternative possibility is that hlh-8 does not depend on SEM-2 at later stages in the M lineage.

Reviewer #2: This manuscript utilizes a novel allele of the SoxC transcription factor, sem-2, isolated in a forward genetic screen. They use a combination of CRISPR genome engineering, traditional transgene reporter analysis and imaging to characterize the phenotype of their sem-2 allele, in the context of the M lineage and the gene regulatory network controlling SM fate, proliferation and differentiation. Their results extend earlier work by their group and others in our understanding of the transcriptional network that underlies the hermaphrodite reproductive musculature.

I have several suggestions where the manuscript could be significantly improved by quantifying images rather than just reporting general trends. The authors did the appropriate quantification for some of their data (Figures 5 & 7), but this quantification is missing from the rest of the manuscript.

Major comments:

- Lines 262-265 – It would be necessary to quantify a series of images acquired using the same settings before concluding that the expression levels are the same between the wild-type and mutant GFP-tagged alleles.

From the micrographs shown in Figure 3 of the endogenous expression and localization of the GFP-tagged SEM-2 allele, it appears that SEM-2 localizes to both the cytosol and nucleus. It might be important to quantify this over developmental time if possible – esp. to see if there are any differences in this localization pattern between the P158S allele. By quantifying this data, the author’s could make a stronger statement, and then the shading in Figure 3L could be a quantitative read-out of the data.

Lines 292-296: In comparing the sem-2::GFP expression in wild-type and sma-9(cc604) backgrounds, it is difficult to assess from a single image – I would recommend quantifying a set of images before making any conclusions about expression levels between different backgrounds.

Figure S2 – I would recommend taking the same approach as you took in Figure 4 for quantifying your data. I would also recommend showing the equivalent expression comparison in panel C/D and then a label showing 20x brighter rather than just putting that information in the figure legend, as it wasn’t immediately obvious to me.

When comparing expression between strains, I would suggest including the quantification rather than just the number of animals examined – for example, in line 357 it states that 100% of mutants displayed reduced expression (100%, N=234).

Instead I would quantify the difference and report it, rather than just the amount of a population that is in the state you are comparing too. This would help with the data and accompanying results in Figure 6 (E-G’’) – I would include quantification of this data, and then report that in the text rather than the % of the population that is indicative of the state (expressing less?

For reporting the proliferation defect in the SM lineage in the sem-2 allele (Figure 8 and accompanying text), I think a summary figure or accompanying plot showing the number of SMs per stage would be appropriate – you could stage animals in sem-2 mutants by VPC divisions.

Discussion –

Line 550-551 – Rather than state that sem-2 might interact with cell cycle regulators interacting with proteins mentioned in a specific review, I would be specific here – possibly through positive regulation of G1/S-phase cyclins and CDKs (Cyclin D/CDK4, Cyclin E/CDK2) or possibly negative regulation of CDK inhibitors like CKIs (CKI-1/CKI-2)?

Minor comments:

Scale bars are missing in the micrographs

For most of the figures, I would recommend showing single color images in grey scale as the human eye can see more details in gray scale than in color. For the overlays, it would be good to avoid red/green combinations as many people are color blind – you could use magenta for red, or any other color-blind friendly look-up-table (LUT) for your overlays. The data was shown this way in some of the figures, but I would recommend doing this for all single channel images, reserving false colors for overlays.

Reviewer #3: 

 Overview: 

This manuscript reports the identification and characterization of a partial loss-of-function allele of the *C. elegans *SoxC transcription factor SEM-2 that was isolated in a screen for suppressors of a *sma-9 *mutant that lacks the development of the coelomocytes in the post-embryonic development of the M lineage. The study has led to a new understanding of the relationship between SEM-2, SMA-9 and LET-381 (a FoxF/C forkhead transcription factor) and of the regulation of another transcription factor in the M lineage, HLH-8. The work has mechanistic implications regarding the transcriptional regulation of cell fate during development and potentially in the pathogenesis of disease since these factors are homologs of proteins that are defective in human developmental disorders. 

General comments: 

This manuscript is well-written and the experimental evidence is convincing. The comments outlines below are fairly minor except for the *hlh-8 *promoter experiment that either needs the approach justified or explained further. 

Specific concerns: 

1. It would be helpful if Fig 1B included some additional information that depicted the relationships between the various proteins described in the introduction and the CC/SM fates that would then be built upon in the Fig 10 model. 

2. The complementation test described in lines 139-141 should define qIs48—is this a GFP reporter (not indicated in the strain table either)? jjIs3900 should be described or explained. And, the isolation of red, non-green animals should be explained as well. 

3. In the Plasmids section of the methods, the purpose of the plasmids should be stated before explaining how the plasmid was made. 

4. In discussing the screen and identification of *jj152 *(lines 217-223), it is confusing how a single allele can be a complementation group. By definition, a group should contain multiple alleles. Also, it should be stated how the *jj152 *phenotypes compare with the original *sma-9 *mutants. 

5. In the results section, it doesn’t make sense how a mutation that is predicted to not touch DNA would affect the DNA binding affinity. Consider rewording or proposing multiple possibilities. 

6. In the experiments where the SoxC binding sites are being investigated in the *hlh-8 *promoter there are several points that need clarifying in both the figures and the text. It is not clear why deletion assays were performed if two putative sites were initially identified. Why weren’t those sites directly mutated once they were noted in the sequence analysis? Was the point of the deletion studies to see if there were any other sites in the region? Or to see whether the regions surrounding where the sites were located were necessary for *hlh-8 *expression? This section of the results needs revising with more explanation. Further, it is not clear in the text or in Fig 10, what kind of mutations are being made to site 1 and site 2. Are they deletions? Are they scrambled sites (which are often better since this would maintain the spacing in the promoter)? Fig 10A should be modified to show the mutations or minimally the nature of the mutations should be stated in the legend. And, why were two site 1 mutant strains made and not a site 2 mutant strain made? Finally, in Fig 10, why are the merged panels not yellow (or at least orange) in B” and E”-G” when there appears to be signal in both of the single channels? 

7. I have several other figure labeling suggestions. In Fig 3 G/H there are bright red cells in addition to the SMs, it would be helpful to have arrows marking the SMs. The legend in Figure 4 is confusing and redundant. Instead of labeling the SMs and SM-like cells white in the upper panels and yellow in the lower panels and describing them twice in the legend, just be consistent with labeling all the panels and define those labels one time in the legend. 

Minor concerns and edits (with line numbers indicated): 

99 and 608: “Coffin-Siris syndrome (CSS)” 

125: “Fiji” should either be explained or have a website indicated 

128: software used for statistical analysis should be indicated 

152 change “give” to “produce” 

153 and throughout manuscript: wild type should be two words and have a dash if modifying a noun such as wild-type animals 

164: the plasmids are listed in Supp Table 3 and not 2 (fix the table numbers in lines 194 and 195 as well) or swap the tables since the text doesn’t match the actual supplementary table numbers. 

166: define IDT 

196: pRF4 not PRF4 

229: indicate the specific transgene (*jjIs1647*?) 

230: Tian et al. 2011 stated here and in the discussion should be numbered like the other references. 

266: say “This prediction is consistent with our genetic…” 

314: after *sma-9 (0) *animals add (Fig 4E-H’’’) 

406: “Similarly, in 50%...” 

419: HLH-8 (8 not in italics) and Fig 7 A-M fix since there is no M in the figure 

421: “in the vms (Fig 7 B, B’). Further, 95% *sem-2*…” 

423: the text states that the mutants were variable but the Fig 7G shows a large standard deviation for WT, too, so needs revising 

435: “At the late L4…” 

562: *arg-1::gfp *should be in italics 

588: P158S should not be in italics 

814: “…Egl phenotype with a uterus filled with late-stage embryos” 

820/822: “A) [Top] Schematic of the …” and “…by an asterisk. [Bottom] Sequence alignment.” 

882: “…mutant animal (L-M”),…” 

897: “images” 

926: WT should not be listed as a *sem-2 *mutant in A,C,E 

940: “C”-H” are merged images” 

954-955: fix the excess italics 

Multiple figure legends have “fluorescence” misspelled. 

In Supp Table 2, last row—pMDB37 “Repair template for generating..” should not be in italics 

**Have all data underlying the figures and results presented in the manuscript been provided?**

Reviewer #1: Yes

Reviewer #2: Yes

Reviewer #3: Yes

PLOS authors have the option to publish the peer review history of their article (what does this mean?). If published, this will include your full peer review and any attached files.

Reviewer #1: No

Reviewer #2: No

Reviewer #3: No

---

## [Decision Letter · Decision Letter 1]

5 Nov 2024

Dear Dr. Liu,

We are pleased to inform you that your manuscript entitled "SEM-2/SoxC regulates multiple aspects of C. elegans postembryonic mesoderm development" has been editorially accepted for publication in PLOS Genetics. Congratulations!

Before your submission can be formally accepted and sent to production you will need to complete our formatting changes, which you will receive in a follow up email. Please be aware that it may take several days for you to receive this email; during this time no action is required by you. Please note: the accept date on your published article will reflect the date of this provisional acceptance, but your manuscript will not be scheduled for publication until the required changes have been made. The reviewers have pointed out some typos and misplaced citations that should also be corrected.

Yours sincerely,

Barth D Grant

Guest Editor

PLOS Genetics

Giovanni Bosco

Section Editor

PLOS Genetics

Aimée Dudley

Editor-in-Chief

PLOS Genetics

Anne Goriely

Editor-in-Chief

PLOS Genetics

Comments from the reviewers (if applicable):

Reviewer's Responses to Questions

**Comments to the Authors:**

Reviewer #1: The authors have addressed all of the reviewers' critiques. The manuscript is improved by the revision. The work clearly advances our understanding of the gene regulatory mechanisms governing development of a mesodermal progenitor cell in C. elegans. It may also be more broadly relevant to human craniofacial development, as the same genes are important for both processes.

Reviewer #2: I commend the authors for integrating all three reviewer comments into the much improved manuscript and appreciate the efforts to present quantified data - I think it strengthens arguments! My only comment is there a typo - "CKD" instead of "CDK" in line 589 p.24.

I would stress the importance of not showing red/green images in your figures - for anyone who is red/green colorblind those overlays and images will be uninterpretable. In ImageJ/FIJI there is a setting to swap out red for magenta - so in theory this would be an easy fix and not require you to remake all your figures.

Reviewer #3: Baccas and colleagues have submitted a substantially revised version of their manuscript with additional experimental data and corresponding text edits. The concerns that I expressed in the initial review have been adequately addressed. I'll note that the file for the revised version did not contain the two Tables found in the initial submission, but I assume the Tables didn't change so I am not concerned about that. On line 582, where the reference has been corrected to be [6], the original reference, Tian et al, 2011, is still in the text and should be deleted. And the new figure legends where "Data is normalized..." is stated should instead read "Data are normalized..." Otherwise, I have no further concerns about the quality of this work.

**Have all data underlying the figures and results presented in the manuscript been provided?**

Reviewer #1: Yes

Reviewer #2: Yes

Reviewer #3: Yes

PLOS authors have the option to publish the peer review history of their article (what does this mean?). If published, this will include your full peer review and any attached files.

Reviewer #1: No

Reviewer #2: No

Reviewer #3: No

**Data Deposition**

http://datadryad.org/submit?journalID=pgenetics&manu=PGENETICS-D-24-00741R1

**Press Queries**

---

## [Editor Report · Acceptance letter]

8 Jan 2025

PGENETICS-D-24-00741R1 

SEM-2/SoxC regulates multiple aspects of C. elegans postembryonic mesoderm development 

Dear Dr Liu, 

We are pleased to inform you that your manuscript entitled "SEM-2/SoxC regulates multiple aspects of C. elegans postembryonic mesoderm development" has been formally accepted for publication in PLOS Genetics! Your manuscript is now with our production department and you will be notified of the publication date in due course.

With kind regards,

Livia Horvath

PLOS Genetics

On behalf of:
